# Application of Perovskite Nanocrystals as Fluorescent Probes in the Detection of Agriculture- and Food-Related Hazardous Substances

**DOI:** 10.3390/polym15132873

**Published:** 2023-06-29

**Authors:** Wei Zhao, Jianguo Zhang, Fanjun Kong, Tengling Ye

**Affiliations:** 1Maize Research Institute, Heilongjiang Academy of Agricultural Sciences, Harbin 150086, China; 2State Key Laboratory of Luminescent Materials and Devices, South China University of Technology, Guangzhou 510640, China; 3Harbin Technician College, Harbin 150500, China; 4School of Chemistry and Chemical Engineering, Harbin Institute of Technology, Harbin 150001, China

**Keywords:** perovskite, nanocrystal, fluorescent probe, agriculture, food

## Abstract

Halide perovskite nanocrystals (PNCs) are a new kind of luminescent material for fluorescent probes. Compared with traditional nanosized luminescent materials, PNCs have better optical properties, such as high fluorescence quantum yield, tunable band gap, low size dependence, narrow emission bandwidth, and so on. Therefore, they have broad application prospects as fluorescent probes in the detection of agriculture- and food-related hazardous substances. In this paper, the structure and basic properties of PNCs are briefly described. The water stabilization methods, such as polymer surface coating, ion doping, surface passivation, etc.; are summarized. The recent advances of PNCs such as fluorescent probes for detecting hazardous substances in the field of agricultural and food are reviewed, and the detection effect and mechanism are discussed and analyzed. Finally, the problems and solutions faced by PNCs as fluorescent probes in agriculture and food were summarized and prospected. It is expected to provide a reference for further application of PNCs as fluorescent probes in agriculture and food.

## 1. Introduction

With the rapid development of Chinese agriculture and food in recent decades, the excessive and indiscriminate use of some hazardous substances (such as pesticides, antibiotics, industrial pollutants, harmful ion pollutants, etc.) is also increasing. As a result, these hazardous substances may cause harm to human beings through agricultural products and food. In order to ensure that the residues of these related hazardous substances do not exceed the standard, fluorescent probes are used as indicators under the excitation of a certain wavelength of light. Through the detection of the generated fluorescence, qualitative or quantitative analysis of the detected substances can be realized. Fluorescent probes play an important role in agriculture and food because of their advantages of simple operation, high sensitivity, and non-destructive visualization [1,2,3].

In the past few decades, fluorescent probes based on semiconductor quantum dots (QDs) have gained wide attention due to their convenience, speed, low cost, and low limit of detection (LOD). To date, a large number of QD-based fluorescent probes have been developed to detect a variety of agricultural- and food-related contaminants, such as pesticides, harmful ion contaminants, chemical dyes, etc. Currently, QDs are generally composed of IV, II-VI, IV-VI, or II-V elements, such as CdTe, ZnO, and ZnS QDs [4]. However, due to the low quantum yield of these QDs, the sensitivity of fluorescent probes prepared by surface modification is low. Therefore, it is necessary to develop new semiconductor fluorescent probes with excellent photoluminescence (PL) performance and high fluorescence quantum yield. Halide PNCs have become a new promising fluorescent probe material in recent years due to their excellent properties, such as high fluorescence quantum yield, tunable band gap, low size dependence, narrow emission bandwidth, and so on [5,6].

In this paper, as shown in Figure 1, the structure and basic properties of halide PNC luminescent materials and the methods to improve the water stability of PNCs are briefly introduced. On this basis, the research progress of PNCs as fluorescent probes for detecting hazardous substances in the field of agricultural and food is reviewed, and the detection effect and mechanism are discussed and analyzed. Finally, the problems and solutions faced by PNCs as fluorescent probes in agriculture and food were summarized and prospected. It is expected to provide a reference for further application of PNCs as fluorescent probes in agricultural and food.

### 1.1. Structure and Basic Properties of PNCs

Halide perovskite is a new type of semiconductor material with the molecular formula ABX_3_, where A mainly represents alkali metal cation cesium (Cs^+^), methylamine ion (MA, CH_3_NH_3_^+^), or formamidine ion (FA, CH(NH_2_)^2+^); B represents metal cations such as Pb^2+^, Sn^2+^, etc.; and X stands for the halide anion (Cl, Br, I, or their combination). As shown in Figure 2, the lattice structure of perovskite, such as CaTiO_3_, has a three-dimensional (3D) angular shared [BX_6_] octahedron, which is formed by the coordination of the B site in the center and six X site anions. Each octahedron is connected through the shared angle to form an extended 3D network structure, while A is embedded in the gap of the network structure [7]. Halide perovskite can have three different crystalline structures: rhombic, tetragonal, and cubic phases. The cubic phase can exist stably at high temperatures [8]. When A is an organic group such as MA or FA, it is called an organic–inorganic hybrid perovskite, whereas when A is Cs^+^, it is called an all-inorganic perovskite.

However, the low exciton binding energy of bulk phase perovskite materials makes carriers easy to be trapped by defects in the lattice, leading to slow radiation recombination of carriers in bulk phase perovskite. To address this problem, efforts have been made to reduce the size of perovskite crystals or reduce its structural dimension to enhance carrier restriction within the perovskite lattice [9,10]. Nano-structured perovskites, such as PNCs, perovskite nanowires, and perovskite nanosheets, can improve the photoluminescent quantum yield (PLQY) of materials due to the quantum domain effect. Among them, PNCs have three dimensions below 100 nm, and perovskite quantum dots (PQDs) have three dimensions below 20 nm, exhibiting a clear quantum limiting effect and high PLQY, which is the most widely studied. The band gap and color of halide perovskite can be continuously adjusted from near-infrared to the violet region by modifying the halide elements and their respective composition, covering the entire visible spectral region. Moreover, the narrow spectral distribution of PNCs is crucial for easy detection in fluorescent probes. Numerous simple and effective synthetic methods are available for obtaining PNCs, such as hot injection [11], room-temperature reprecipitation [12], solvothermal synthesis [13], microwave method [14], and ultrasonication [15], among others.

### 1.2. Methods to Improve the Stability of PNCs

Although PNCs possess excellent properties, as ionic crystals they have a large surface energy and are susceptible to external influences, leading to instability and degradation of their structures. For instance, water, light radiation, and temperature changes can all cause fluorescence quenching and structural degradation [16]. In applications such as agriculture and food, where hazardous substance detection requires a water environment, stability in water is crucial for PNCs to function effectively as fluorescent probes. Fortunately, recent studies have developed a range of strategies to address the limitations of PNCs, including surface coating, metal ion doping, and surface passivation.

#### 1.2.1. Surface Coating by Polymer, Inorganic Oxide, and Porous Materials

The surface of the PNCs is coated and isolated from the environment, which greatly improves the stability of the PNCs, especially the water resistance. At present, PNCs are coated with polymer, inorganic oxide, and porous materials. The polymer forms a dense network on the NC surface, which can effectively prevent environmental erosion. Researchers usually dissolve or expand the polymer in toluene, N, N-dimethylformamide, and other solvents, realize the encapsulation protection of PNCs through in situ growth or adsorption, and then remove the solvent under vacuum and high temperature to obtain polymer–PNCs composite materials [17,18]. As shown in Figure 3, Xuan et al. used divinylbenzene, ethyl acetate, and azodiisobutyronitrile (AIBN) as reaction raw materials to prepare superhydrophobic organic polymers through solvothermal methods. High-quality PQDs were prepared by hot injection, and the cyclohexane solution of PQDs was mixed with the organic polymer to fully absorb the QDs into the pores on the surface of the polymer, forming a complex (CPB@SHFW). Fluorescence quantum efficiency remained at 91% after immersion in water for one month, and the contact angle of the complex was maintained at 150°. The pores of the polymer provided a superhydrophobic environment to improve the water stability of the QDs, while also acting as a framework to maintain the excellent luminous performance of PQDs. This study successfully achieved the preparation of “water-resistant” perovskite QDs [19].

Although the hydrophobic strategy used to coat PNCs has improved their stability, the application of fluorescent probes often requires PNCs to be dispersed in a water environment. To address this challenge, the introduction of amphiphilic polymers that are both hydrophilic and hydrophobic has been shown to be an effective solution. Avugadda et al. reported the creation of CsPbBr_3_ NCs in capsules that maintain a PLQY of approximately 60% for over two years in water, using a few scalable fabrication steps that involve an automated routine. For this purpose, an encapsulating amphiphilic polymer, Polystyrene-block-poly(acrylic acid) (PS-b-PAA), with low molecular weight (34 kDa) was selected. As illustrated in Figure 4, this polymer, in the presence of NCs in toluene and with the addition of methanol (MeOH) in a single-phase system, facilitates the formation of polymer capsules containing the NCs. This method does not require prior surface modification of the as-synthesized NCs and is insensitive to their surface coating. CsPbBr_3_@PS-b-PAA is soluble in water, and its emission remains stable in a saline solution [20].

Compared to polymers, inorganic nano oxide coatings can also play a role in stabilizing PNCs, with the advantage of better water dispersion. Nano SiO_2_ has become a typical representative of inorganic materials used for coating PNCs due to its high heat resistance, stability, and transparency. Li et al. have developed a simple method for producing water-resistant PQDs@SiO_2_ nanodots (wr-PNDs) that can be easily dispersed in water and maintain their emission for up to six weeks, demonstrating an unprecedented water-resistance capability. The synthetic process for creating wr-PNDs is illustrated in Figure 5. Initially, pristine PQDs are dispersed in a mixed solvent of toluene and (3-mercaptopropyl)trimethoxysilane (C_6_H_15_O_3_Si-SH, MPTMS), followed by the addition of de-ionized water to initiate the hydrolysis of MPTMS. MPTMS molecules can be tightly adsorbed onto the PQDs due to the formation of Pb-S bonding. These bound molecules reduce the interfacial energy between the PQDs and silica by providing additional steric stabilization, thereby preventing phase separation between the PQDs and silica. Silica-encapsulated PQDs-Pb-S-SiO_2_-SH nanodots are produced as hydrolysis and condensation of MPTMS are triggered by the added water [21].

Porous materials provide a high degree of porosity that can be used to prevent self-aggregation of PNCs and reduce environmental erosion, making it another viable method for enhancing the stability of PNCs coated with inorganic materials. A universal synthesis method of PNCs with pore domain limiting-shell isolation dual protection was developed by You et al.; as depicted in Figure 6. In this approach, perovskite precursor was injected into mesoporous silicon, and PNCs were grown in situ in the mesoporous silicon pores through heating and annealing. Next, dense Al_2_O_3_ shells were deposited on the mesoporous silicon surface through atomic deposition technology, which effectively isolates the impact of the environment on PNCs, owing to the abundant -OH on the mesoporous silicon surface. The resulting mSiO_2_-CsPbBr_3_@AlO_x_NCs can remain stably dispersed in water for more than 90 days and can effectively shield the ion exchange effect, thereby significantly enhancing the stability of PNCs and expanding their practical application range [22]. Likewise, Zhang et al. synthesized ceramic-like stable and highly luminescent CsPbBr_3_ NCs by embedding CsPbBr_3_ in silica derived from molecular sieve templates at high temperature (600–900 °C). The resulting CsPbBr_3_-SiO_2_ powder exhibits high PLQY (~71%) and excellent stability, which can be compared with the ceramic Sr_2_SiO_4_:Eu^2+^ green phosphor [23].

Metal–organic framework compounds (MOFs) are a novel class of porous materials that are assembled by inorganic metals and organic ligands. Due to their high porosity and large specific surface area, MOFs are often utilized as carriers for nanoparticles, drugs, and other materials. Incorporating PNCs into the pores of MOFs can significantly enhance their stability. However, in order to function as a carrier for PNCs, the MOF design must address several challenges. Firstly, the MOF matrix must remain stable during PNCs incorporation and fluorescence detection. As many bivalent metal-based MOFs are sensitive to water or ambient moisture, high-valent metal-based MOFs are ideal hosts that provide stable pore structures for encapsulation. Secondly, the aperture of MOFs must be sufficiently large to accommodate PNCs. While micropore MOFs with apertures smaller than 2 nm are frequently used as encapsulation materials for perovskite nanoparticles, the limited space can hinder QD growth and damage the micropore structure. Therefore, mesoporous MOFs with pore sizes greater than 4 nm are more suitable for PNC fixation. In the case of CsPbBr_3_ nanocrystals, QDs with complete crystal structures are more suitable for fluorescence detection applications to ensure efficient light absorption and prevent dense surface recombination. Simultaneously, the mesoporous cage can effectively separate the QDs and realize the quantum limited domain of perovskite. As illustrated in Figure 7, Qiao et al. encapsulated CsPbBr_3_ nanocrystals into a stable ferric organic skeleton (MOF) with mesoporous cages (5.5 and 4.2 nm) via a sequential deposition pathway to obtain a perovskite–MOF composite CSPbBr_3_@PN-333 (Fe). The CsPbBr_3_ nanocrystals are stabilized by the restriction of the MOF cage without aggregation or leaching [24]. Wu et al. employed a sequential deposition method to encapsulate MAPbI_3_ in cheap iron-based MOFs (PSN-221) to obtain the composite MAPbI_3_@PCN-221(Fe_x_) (x = 0–1), which they used for photocatalytic CO_2_ reduction. Due to the protection offered by PSN-221 (Fe_x_), this composite catalytic system exhibits ultra-high stability and can operate continuously in water for more than 80 h. Additionally, the maximum efficiency of photocatalytic CO_2_ is 38 times higher than that of pure PCN-221(Fe_x_) [25].

#### 1.2.2. Ion Doping

Ion doping is a widely used and effective method for introducing heteroatoms into the lattice, making it one of the most important approaches for modifying semiconductors. This method is valuable because it can maintain the structure and properties of the original main crystal to a large extent. Perovskite structures have a high tolerance for impurity doping, enabling various impurity atoms to be doped into the lattice of PQDs. This doping strategy can significantly improve the stability of perovskite and regulate its photoelectric characteristics [26]. The typical perovskite structure (ABX_3_) consists of corner-sharing [BX_6_]^4−^ octahedra, with *A*-site cations confined within the cuboctahedra cages. In the ABX_3_ structure, the ionic radii of *A*-site, *B*-site, and *X*-site ions must correspond to the Goldschmidt tolerance factor *t* [27]:$$t=\frac{r_{A}+r_{X}}{\sqrt{2}\left(r_{B}+r_{X}\right)}$$

The Goldschmidt tolerance factor is defined by the formula *t* = (*r_A_* + *r_X_*)/√2(*r_B_* + *r_X_*), where *r_A_*, *r_B_*, and *r_X_* are the ionic radii of the *A*-site, *B*-site, and *X*-site elements, respectively [26]. To maintain a stable perovskite structure, t should fall between 0.813 and 1.107, with a range of 0.9 ≤ *t* ≤ 1 generally considered optimal for the cubic phase [7]. The [BX_6_]^4−^ framework confines the *A*-site cations, and those that are either too large or too small can cause the cubic-like crystal structure to distort, warp, and eventually break down. MA (2.70 Å) is generally the most suitable size for the cubic perovskite structure, whereas FA (2.79 Å) and Cs (1.88 Å) are slightly too large or small to serve as ideal *A*-sites [28], leading to deviations from the equilibrium interplanar distances. *A* mixed *A*-site cation system that combines larger and smaller ions can help to stabilize the 3D perovskite framework and overcome these issues. For instance, Protesescu et al. replaced some Cs^+^ ions with FA^+^ ions to improve the structural stability of CsPbI_3_ PQDs, resulting in a higher t value relative to pure CsPbI_3_ PQDs. The resulting FA_0.1_Cs_0.9_PbI_3_ PQDs display a 3D orthogonal structure, stable red-light emission (685 nm), a PLQY over 70%, and can maintain their PLQY in solution (with less than 5% relative decrease) after several months of storage at ambient conditions, owing to the restricted transformation from 3D cubic perovskite structure to 1D polycrystals [29]. Triple cations such as (Cs, FA)PbI_3_ or (Cs, MA, FA)PbI_3_ have also demonstrated improved stability by promoting phase structure stability, and multi-cationic doping of *A* sites has emerged as a key method for obtaining high-quality perovskite films [30]. However, it is rarely reported in the field of PNCs [31]. In addition, Wang et al. synthesized ultra-stable quasi-2D PQDs by introducing butadiamine cation (BA^+^) into methylamine lead bromide perovskite (MAPbBr_3_), as shown in Figure 8. The quasi-2D perovskite (BA)_2_(MA)_x−1_Pb_x_Br_3x+1_, which has reduced dimensionality, exhibits higher luminous efficiency and better environmental stability than traditional 3D perovskite due to the higher exciton binding energy and formation energy of the reduced dimension perovskite [32].

The incorporation of dopants to replace B-site Pb^2+^ in perovskite structures has been shown to significantly enhance their stability or modify their optical properties, examples of which include Zn^2+^, Ni^2+^, and RE ions [33,34]. The smaller ionic radius of Zn^2+^ (0.74 Å) compared to Pb^2+^ (1.19 Å) causes the perovskite lattice to contract, leading to an increase in exciton binding energy and promoting radiative recombination within CsPbI_3_ PQDs [35]. In addition, Mn^2+^ doping also increases the Goldschmidt tolerance factor (t), enhancing structural stability [36]. Similarly, the introduction of Ni^2+^ ions into CsPbI_3_ PQDs results in lattice contraction due to the smaller ionic radius of Ni^2+^ (≈0.69 Å) relative to Pb^2+^ (≈1.19 Å), resulting in a blue shift in the PL and a higher quantum efficiency, as shown in Figure 9a,b. Moreover, Ni^2+^ doping improves the stability of PNCs by increasing t relative to CsPbI_3_ PQDs, leading to a more stable perovskite structure. The stability of Ni:CsPbBr_3_ PNCs with distinct Ni/Pb ratios against moisture and UV light was tested and shown in Figure 9c,d, providing compelling evidence for the beneficial effects of Ni^2+^ dopants on emission performance [37].

#### 1.2.3. Surface Passivation

The large specific surface area of PNCs results in a high concentration of surface atoms, which in turn leads to an abundance of dangling bonds and defects. These defects can trap electrons in the conduction band and cause non-radiative energy loss, thereby reducing the stability of nanocrystals. Surface passivation using appropriate ligands is an effective strategy to improve the stability of PNCs. However, commonly used ligands like oleic acid and oleylamine are dynamic and can easily desorb in the purification process, leading to PNC aggregation and loss of optical properties. Thus, ligands that bind more firmly to the surface of PNCs are required for better passivation, reducing the impact of water and oxygen and improving stability. Polydentate ligands, containing multiple functional groups, are shown to bind more strongly to nanocrystals and provide improved stability. For instance, Figure 10 illustrates a strategy for introducing polydentate ligand of AHDA to synthesis stable PNCs. The polyamine chelating ligand N′-(2-aminoethyl)-N′-hexadecylethane-1, 2-diamine (AHDA) can anchor the PNC surface lattice with a higher binding energy (2.36 eV) than that of the commonly used oleylammonium ligands (1.47 eV). The strong chelation effect achieves ultra-stable CsPbI_3_ PNCs with a high binding energy, exhibiting excellent stability even under harsh conditions such as repeated purification, polar solvents, heat, and light [38]. The use of mercaptan materials such as 2-amino-ethylmercaptan (AET) can also lead to tightly bound ligand layers on the surface of PNCs, resulting in high stability, even in water. The AET-CsPbI_3_ QDs (in both solutions and films) retained more than 95% of the initial PL intensity in water after 1 h [39]. These strong chelating ligand strategies offer promising avenues for achieving stable and high-performance PNCs for a range of applications.

In recent years, zwitterionic organic ligands have emerged as a promising approach for surface engineering of PNCs and defect elimination, owing to their multifunctional zwitterionic structure. As shown in Figure 11, Krieg et al. proposed a surface passivation strategy for PQDs using long-chain zwitterionic ligands such as sulfobetaine and phosphocholine, which comprise both cationic and anionic groups. These zwitterionic ligands can coordinate with both surface cations and anions and are firmly attached to the surface of PQDs through the chelation effect, thereby obtaining PQDs with high stability. For example, 3-(N,Ndimethyloctadecylammonio) propanesulfonate can result in much improved chemical durability. The sulfobetaine-capped CsPbBr_3_ NCs retained PLQYs in the range of 70–90% for 28–50 days under ambient conditions [40]. Moreover, they demonstrated that the strong binding between the amphoteric lecithin and PQDs provides CsPbBr_3_ PQDs solutions with superior colloidal stability over a wide range of concentrations [41]. In another study, Wang et al. used polymers containing multiple sulfobetaine amphoteric groups to achieve desirable surface coatings and CsPbBr_3_ PQDs that maintain bright PL emission even after 1.5 years of storage in polar solvents such as acetone and ethanol [42].

In addition, the presence of long alkyl chain organic ligands in PQD films severely impedes charge transport and interlayer coupling, calling for the development of more conductive organic semiconductor ligands to prepare PNCs with favorable optoelectrical properties. Pan et al. demonstrated that the incorporation of organic semiconductor molecules (rhodamine B derivative, COM) into CsPbBr_3_ NCs resulted in COM-CsPbBr_3_ NCs with a high PLQY of 82% and exceptional stability under harsh commercial accelerated operating stability tests such as high temperature (85 °C) and high humidity (85%). The product exhibits remarkable endurance against high temperature and high humidity, retaining 84% of the initial PL intensity value for 300 h. The stability of CsPbBr_3_ NCs can be significantly improved by the strong interaction between organic semiconductor molecules and the quasi-type II heterostructure formed by CsPbBr_3_ and moderate photocarrier transfer. Furthermore, the diversity and versatility of organic semiconductors present a broad prospect for their combined application in the field of luminescence and display [43].

Table 1 summarizes methods to improve the stability of PNCs and their stability performance. Surface coating is a way from outside to inside, playing the main role of encapsulation. Metal ion doping and surface passivation are ways to improve the intrinsic structural stability from inside to outside. They are not absolute by classification and may work together to improve the stability of PNCs sometimes.

## 2. Application of PNCs as Fluorescent Probes in the Detection of Agriculture- and Food-Related Hazardous Substances

### 2.1. The Detection of Agriculture-Related Hazardous Substances

#### 2.1.1. Pesticide Residue Detection

Pesticides are crucial in minimizing crop losses, ensuring food security, preventing the emergence and spread of plant diseases and insect pests, and improving health conditions. However, their use also leads to environmental pollution, pesticide residues, and other issues. Pesticide residues refer to the presence of trace amounts of pesticide precursor, toxic metabolites, degradants, and impurities that remain in grains, vegetables, fruits, livestock products, aquatic products, soil, and water bodies even after the application of pesticides in agricultural production. Currently, the annual production of chemical pesticides worldwide is nearly 4.1 million tons, and these synthetic compounds were utilized as pesticides, fungicides, algaecides, deworming agents, defoliants, and other pesticides [44]. The substantial use of pesticides, particularly organic pesticides, has resulted in severe pesticide pollution. Ingesting food containing high levels of highly toxic pesticide residues can lead to acute poisoning in both humans and animals. While long-term consumption of agricultural products with excessive pesticide residues may not cause acute poisoning, it can result in chronic poisoning in humans, leading to the development of diseases, inducing cancer, and even affecting future generations [45]. Therefore, the real-time and rapid detection of pesticide residues in agricultural products is of great significance for global food safety and environmental safety. Researchers have conducted significant research on rapid detection technologies for pesticide residues, especially for organophosphorus, organochlorine, triazine herbicides, and other pesticides, yielding some remarkable achievements.

##### Organophosphorus Pesticide

Organophosphorus pesticides are a type of pesticide that contains organic compounds with phosphorus. They are widely used in agricultural production to control plant diseases and insect pests but can leave residues in crops. Ensuring food safety requires rapid and accurate detection of organophosphorus pesticides. 

Dimethoate oxide (OMT) is an organophosphorus insecticide with a strong imbibition insecticidal effect. It can be absorbed into the plant stem and leaves and transmitted to other parts of the plant to kill harmful pests. Huang et al. successfully synthesized a new MIPs@CsPbBr_3_ QDs composite using imprinting technology with a sol–gel reaction, as shown in Figure 12. APTES-capped CsPbBr_3_ QDs were used as a fluorescent carrier, and TMOS was used as the cross-linker to gradually form a silica matrix, improving the stability of perovskites, and synthesized water-soluble PQDs. OMT was selected as the template molecule due to its hydrogen bonding interaction with APTES. Loading the MIPs with OMT significantly quenched the fluorescence of MIPs@CsPbBr_3_ QDs, with a linear range of OMT from 50 to 400 ng/mL and a LOD of 18.8 ng/mL. After removing the OMT template via solvent extraction, the resulting MIP@CsPbBr_3_ QD composites showed selective recognition ability towards specific template-shaped molecules such as dimethoate, with an imprinting factor of 3.2, indicating excellent specificity of the MIPs for inorganic metal halide (IMH) perovskites [4].

2,2-dichlorovinyl dimethyl phosphate (DDVP) is a highly effective and wide-spectrum organophosphorus insecticide commonly used to control various pests on crops such as cotton, fruit trees, vegetables, sugarcane, tobacco, tea, and mulberry. However, on 27 October 2017, the World Health Organization’s International Agency for Research on Cancer classified DDVP as a group 2B carcinogen. In a study by Huang et al.; molecularly imprinted mesoporous silica (MIMS) with in situ perovskite CsPbBr_3_ quantum dots (QDs) was prepared and applied for the fluorescence detection of DDVP, as shown in Figure 13. The QDs stability was greatly improved by encapsulating them in mesoporous silica, resulting in higher analytical performance. The design of PQDs-encapsulated MIMS is based on PQDs being encapsulated into mesoporous silica and acting as a transducer for recognition signal fluorescent readout, while molecular imprinting on the surface of mesoporous silica provides selectivity. CsPbBr_3_ QDs with uniform size and high fluorescence intensity were first grown directly in the channels of mesoporous silica SBA-15. Then, molecular imprinting was applied to the surface of QDs-encapsulated mesoporous silica using a sol–gel method. The mesoporous silica provided a solid support for molecular imprinting and acted as a medium for the in situ synthesis of QDs. The QDs’ fluorescent properties were expected to be affected once the template molecules were captured in the binding sites, reducing the distance between the template and the QDs. Under optimized conditions, the QDs-encapsulated MIMS showed a linear relationship between DDVP concentration of 5–25 μg/L, and the LOD was 1.27 μg/L. The recovery of DDVP increased from 87.4% to 101% [46].

Phoxim is a highly effective broad-spectrum organophosphorus insecticide, exhibiting properties of stomach toxicity, contact pesticide, and long-lasting effect under dark conditions. To detect phoxim specifically and sensitively in samples, Tan et al. synthesized perovskite CsPbBr_3_ QDs embedded in molecularly imprinted polymers (MIP) by thermal injection. As depicted in Figure 14, CsPbBr_3_ QDs with high-brightness cores were encapsulated by MIPs using a sol–gel approach initiated by trace moisture in the air. To overcome the disadvantage of using siloxane monomers with simple organic groups, a multifunctional monomer, BUPTEOS, was synthesized with phenyl and hydrogen bond donors. During pre-polymerization, BUPTEOS and phoxim formed a complex through hydrogen bond interactions and π-π accumulation, leading to the formation of specific imprinted cavities in the MIP/QDs composites upon slow sol-gel molecular imprinting process and removal of templates. The resulting MIP/QDs complex demonstrated excellent selectivity to phoxim, with a blotting factor of 3.27. Compared to previous studies on detection of organophosphorus pesticides, the MIP/QDs fluorescent probe exhibited high sensitivity and specificity. Under optimized conditions, the fluorescence quenching of MIP/QDs showed a good linear correlation with the concentration range of 5–100 ng/mL, and the LOD was 1.45 ng/mL. By encapsulating the QDs of the novel MIP/perovskite CsPbBr_3_ QDs fluorescent probe in the imprinted silica matrix, the stability of the QDs in phoxim detection was significantly improved. In this study, siloxane functional monomers with two functional groups were synthesized by in situ slow hydrolysis of organosilicon monomers [47].

##### Triazine Herbicides

Triazine herbicides can be classified into symmetrical triazines (1,3,5-triazine) and asymmetric triazines (1,2,4-triazine). The main commercial herbicide is symmetrical triazine, which can be further divided into thiomethyl-s-triazine (e.g., prometryn, ametryn, and terbutryn), simazine, terbuthylazine, fluoroalkyl-s-triazine (e.g., indaziflam and triaziflam), and methoxytriazine (e.g., prometon and terbumeton) [48].

Prometryn is commonly used for pre- and post-bud weeding in cotton and bean fields as well as for algae removal in aquatic products. Zhang et al. developed a novel molecularly imprinted electrochemical luminescence (MIECL) sensor by using perovskite quantum dots (PQDs) coated with a molecularly imprinted silica layer (MIP/CsPbBr_3_-QDs) as recognition and response elements. As shown in Figure 15, the MIP/CsPbBr_3_-QDs layer was immobilized onto the surface of a glassy carbon electrode (GCE) via one-pot electropolymerization to enhance water resistance, improve the stability of CsPbBr_3_-QDs, and reduce film shedding. The highly selective and ultrasensitive MIECL sensor was successfully constructed for prometryn analysis in fish and seawater samples. The LODs for fish and seawater samples were 0.010 μg/kg and 0.050 μg/L, respectively. The recoveries of fish and seawater samples were 88.0% to 106.0%, with a relative standard deviation lower than 4.2%. The MIECL sensor based on MIP/CsPbBr_3_-QDs shows good stability, accuracy, and precision and can sensitively detect prometryn in aquatic products and environmental samples [49].

Simazine is a highly effective selective herbicide that has been listed as a category 3 carcinogen by the International Agency for Research on Cancer (IARC) on 27 October 2017. To detect simazine in aquatic products, Pan et al. developed a novel MIECL sensor based on the molecular-imprinted polymer perovskite (MIP-CsPbBr_3_) and Ru(bpy)_3_^2+^ luminescent molecules. Under optimal experimental conditions, MIP-CsPbBr_3_-GCEs were exposed to different concentrations of simazine solution (0.1–500 μg/L) and analyzed using the MIECL sensor. The ECL strength showed a linear relationship with the logarithm of simazine concentration within the range of 0.1–500.0 μg/L, and the limit of detection (LOD) was 0.06 μg/L. The MIECL sensor method was validated by analyzing fish and shrimp samples with recovery rates of 86.5–103.9% and relative standard deviations lower than 1.6%. The developed MIECL sensor demonstrated excellent selectivity, sensitivity, reproducibility, accuracy, and precision for the determination of simazine in actual aquatic samples [50].

##### Organochlorine Pesticides

Organochlorine pesticides are a group of organic compounds containing organochlorine elements, commonly used to prevent and control plant diseases and insect pests. They can be categorized into two main types: benzene-based and cyclopentadiene-based. Yang et al. demonstrated that CH_3_NH_3_PbBr_3_ quantum dots (MAPB-QDs) undergo a blue shift in fluorescence spectra upon exposure to polar organochlorine pesticides (OCPs), providing a method for the detection of these compounds. The presence of polar OCPs alters the structure and emission properties of MAPB-QDs by desorbing the capping ligands OA and OAm from their surface, resulting in defect sites that can be occupied by polar OCPs. This allows the chlorine element in OCPs to fully dope into the QDs, increasing the band gap of the MAPB-QDs and causing a blue shift in their wavelength. To address the insufficient stability of MAPB-QDs in the presence of moisture, the researchers mixed MAPB-QDs with PDMS to create a colorimetric card that can be used for the rapid detection of OCPs in actual samples. This study represents the first use of MAPB-QDs for the detection of polar OCPs (Figure 16) [51].

##### Other Pesticides

Clodinafop is a herbicide belonging to the aryloxy phenoxy propionate class, soluble in various organic solvents but decomposes under strong acidic or basic conditions. Vajubhai et al. synthesized CsPbI_3_ PQDs using a microwave radiation method and utilized them for detecting clodinafop in samples via fluorescence spectrometry. Figure 17 illustrates the simple synthesis method of CsPbI_3_ PQDs utilizing PbI_2_, Cs_2_CO_3_, OAm, OA, and ODE as reagents. The resulting PQDs showed a strong red emission at 686 nm and were well-dispersed in hexane. They were utilized as a probe for fluorescence detection of clodinafop in an organic phase (hexane), where the strong red emission at 686 nm was quenched by clodinafop. The study found a good linear relationship between the quenching intensity and the concentration of clodinafop in the range of 0.1–100 μM, with a LOD of 34.70 nM. Furthermore, the fluorescent probes were integrated into liquid–liquid microextraction for the fluorescence analysis of clodinafop in fruits, vegetables, and grains. The developed probes showed a good linearity to the concentration of clodinafop in the range of 0.1−5.0 μM, with a LOD of 34.70 nM, and exhibited a good recovery rate (97−100%) and low relative standard deviation. The results indicate that CsPbI_3_ PQDs can be used as a sensor for the quantitative detection of clodinafop in samples [1].

Propanil is a highly selective contact herbicide that contains amides, primarily used in rice fields or seedling fields to control barnyard grass and other gramineous and dicotyledonous weeds. The MIP-QDs, under optimized parameters, showed good linearity in the concentration range of paspalum from 1.0 μg/L to 2.0 × 10^4^ μg/L. The developed MIP-QD-based fluorescent probe exhibited a good recovery rate ranging from 87.2% to 112.2%, with relative standard deviations of less than 6.0% for fish and seawater samples. The LOD of paspalum in fish and seawater was 0.42 μg/kg and 0.38 μg/L, respectively. Fluorescence test strips based on MIP-QD also showed satisfactory recovery rate ranging from 90.1% to 111.1%, and the LOD of paspalum in seawater samples was 0.6 μg/L. The developed fluorescent probe and test strip were successfully used for the detection of paspalum in environmental and aquatic products [52].

As shown in Table 2, PNCs demonstrate good stability, high sensitivity, low LOD, and high recovery rate for conventional pesticides. Furthermore, they can be developed into test strips for rapid detection with broad and convenient application prospects.

#### 2.1.2. Environmental Pollutant Detection

Uranium is a radioactive element found in various chemical forms in the environment, with uranyl ion (UO_2_^2+^) being the most common valence state under normal conditions. Inhalation or ingestion of U can lead to irreversible kidney damage, destruction of biomolecules, and DNA damage. According to World Health Organization guidelines, the concentration of UO_2_^2+^ in drinking water should be less than 30 ppb. As most uranium complexes are highly soluble in water and mobile in aquatic environments, detecting its presence is essential for monitoring agroecological systems.

Halali et al. synthesized all-inorganic CsPbBr_3_ PQDs with bright PL through thermal injection and used them for UO_2_^2+^ sensing. As shown in Figure 18, the first step involved the adsorption of UO_2_^2+^ on CsPbBr_3_ PQDs, and zeta potential measurements confirmed the process of electrostatic attraction and adsorption, facilitating quenching. Dynamic light scattering (DLS) revealed that CsPbBr_3_ PQDs treated with uranyl ions had significantly larger hydrodynamic radii (1244 nm) than the untreated probe (47.9 nm), indicating adsorption of UO_2_^2+^ on CsPbBr_3_ PQDs. Fluorescence quenching resulted from the adsorbed UO_2_^2+^ ions on the surface of CsPbBr_3_ PQDs. The ultra-low LOD of the probe was 83.33 nM (19.83 ppb), and it demonstrated rapid detection of UO_2_^2+^ even in non-polar solvents such as toluene. This study suggests new avenues for designing probes (such as PQDs) and analytes (such as metal ions) for trace level sensing of metal ions [53].

O-nitrophenol (ONP) is a member of the phenolic compound family and is widely used as an intermediate in organic synthesis, dyes, medicines, and other products. ONP is also employed as a photographic developer, pH indicator, and photosensitive material, and as a rubber antioxidant. However, ONP is highly toxic and is considered a priority pollutant for environmental analysis and monitoring. Inhaling ONP can cause adverse effects such as headache, dizziness, and nausea, and serious exposure can lead to damage to the central nervous system or the liver/kidney function. Deng et al. developed a molecularly imprinted fluorescent probe using a combination of highly luminescent PQDs and molecular imprinting technology to rapidly detect ONP (see Figure 19). To enhance the hydrophobic properties of the PQDs, a superhydrophobic porous organic polymer framework (SHFW) was created by combining a superhydrophobic material with a pore-forming agent. This SHFW protected the PQDs from contact with water vapor in the air and increased their stability. The CsPbBr_3_@SHFW fluorescent probe was designed using SHFW-modified CsPbBr_3_ QDs to detect ONP in river silt. The probe exhibited high selectivity for ONP, and under optimal detection conditions, the linear range of CsPbBr_3_@SHFW for ONP detection was 0–280 μM, with a limit of detection of 7.69 × 10^−3^ μM [54].

Table 3 summarizes the PNC-based fluorescent probes for environmental pollutant detection. High fluorescence performance of CsPbBr_3_ PQDs proved them to be efficient fluorescent probes for the detection of pollutants U and ONP.

#### 2.1.3. Ion Detection

##### Cation

As industrialization and urbanization continue to progress, harmful ion pollutants have become a major source of land and water pollution, with highly toxic ions posing a global concern. Conventional detection methods are beset with issues such as high cost, long detection cycle, low accuracy, and low efficiency. Thus, the development of simple and selective sensors is crucial for detecting harmful ion pollutants. Compared to other methods, the fluorescence-based method has advantages such as low cost, high sensitivity, fast detection, and ease of use. Using PNCs to develop trace detection methods represents a breakthrough in new methods.

Liu et al. synthesized all-inorganic CsPbBr_3_ PQDs via the thermal injection method as a fluorescent probe for the detection of Cu^2+^ in organic phase. The cubic-phase CsPbBr_3_-NC exhibits a quantum yield as high as 90%. The PL intensity of CsPbBr_3_ PQD was quenched significantly within seconds of adding Cu^2+^ owing to effective electron transfer from PQD to Cu^2+^, as revealed by absorption spectra and transient PL lifetime experiments. The sensor detected Cu^2+^ in the range of 0 to 100 nM, with a low LOD of 0.1 nM, demonstrating excellent sensitivity and selectivity for Cu^2+^ in hexane [55].

Li et al. established a method for detecting Cu^2+^ in aqueous solution based on CsPbBr_3_ (CPB) QDs. A strong organic ligand (oleylamine, OAm) was added, and due to the strong interaction between the amine ligand and metal ions, OAm in cyclohexane captured Cu^2+^ from water, forming an OAm-Cu^2+^ complex at the cyclohexane/water interface (Figure 20). These Cu^2+^ complexes rapidly diffused into cyclohexane, ultimately quenching the PL of CPB QDs. The fluorescence probe based on CPB QDs exhibited a wide linear range (106 M–102 M), a short response time (1 min), and good selectivity for Cu^2+^. The metal ion phase transfer strategy avoided direct application of unstable PQDs in a water environment [56].

Mercury ion pollution is a significant issue that impacts human health and environmental safety. To address this problem, Lu et al. developed a novel fluorescence nanosensor based on the surface ion exchange mechanism for rapid, highly sensitive, and selective visual detection of mercury ions (Hg^2+^), using CH_3_NH_3_PbBr_3_ PQDs. CH_3_NH_3_PbBr_3_ PQDs were synthesized using ligand-assisted reprecipitation (LARP) technology, without adding any modifier. These PQDs exhibit strong green fluorescence with a high quantum yield of 50.28%. The main principle of the sensor is based on the fact that the strong green fluorescence of CH_3_NH_3_PbBr_3_ QDs at 520 nm is significantly quenched by Hg^2+^, and the blue shift occurs with the increase of Hg^2+^ fluorescence due to surface ion exchange, which differs from most other detection mechanisms shown in Figure 21. The LOD of the sensor is 0.124 nM (24.87 ppt) within the range of 0 nM to 100 nM. The fluorescence intensity of PQDs remains unaffected by interfering ions such as Cd^2+^, Pb^2+^, Cu^2+^, etc.; indicating high selectivity of the fluorescent probe towards Hg^2+^. The detection of Hg^2+^ using CH_3_NH_3_PbBr_3_ PQDs has several advantages, including simple operation, small sample size, high efficiency, and visual detection [57].

Chen et al. developed a novel method for detecting Fe(III) by encapsulating CsPbBr_3_ PQDs into poly(styrene/acrylamide) nanospheres using an improved expansion and contraction strategy, as illustrated in Figure 22. The CPB PQDs were dissolved in toluene, which also served as a swelling agent for PSAA polymers, while isopropanol was used as a dispersant for PSAA spheres. After ultrasonic swelling treatment, the composites were rapidly extracted from n-hexane. The resulting CPB@PSAA composites were well-dispersed in an aqueous solution and maintained their bright fluorescence for over 12 months. The fluorescence of the solution could be selectively quenched by the presence of iron ions, enabling the selective sensing of Fe(III). The results demonstrated a good linear relationship between 5 and 150 μM under optimal conditions, with an LOD of approximately 2.2 μM. Moreover, Fe(III) sensing was successfully applied to Yangtze River water, human serum, and tea with satisfactory results [58].

In 2019, Wang et al. developed a novel composite material, carbon quantum dots (CQDs) doped MAPbBr_3_ PQDs encapsulated with ultra-stable SiO_2_, for potential applications in fluorescence analysis. The composite material was prepared by using carboxy-rich CQDs and MAPbBr_3_ to form hydrogen-bond interactions, as depicted in Figure 23, which significantly improved the thermal stability of CQDs-MAPbBr_3_. Furthermore, the composite material was encapsulated with highly dense SiO_2_ via in situ growth, which greatly enhanced its water resistance and stability. The composite material exhibited excellent water and thermal stability for over 9 months in aqueous solution and maintained strong fluorescence emission even after annealing at 150 °C. The composite material was then employed for fluorescence analysis to detect Zn^2+^ and Ag^+^ [59].

Yan et al. developed a novel fluorescence detection platform for selective detection of Pb^2+^ ions based on the excellent luminescence properties of lead halide perovskite CH_3_NH_3_PbBr_3_ (MAPbBr_3_) and CsPbBr_3_. As shown in Figure 24, a high concentration of CH_3_NH_3_Br (MABr) solution was employed as a fluorescent probe. Upon adding PbBr_2_, a rapid chemical reaction occurred to form MAPbBr_3_, which exhibited a significant luminous response under ultraviolet light. The fluorescence probe mechanism for Pb^2+^ is attributed to the excellent PL properties of MAPbBr_3_ in MAPbBr_3_@MABr solution. Hence, the non-fluorescent MABr demonstrated a sensitive and selective luminescence response to Pb^2+^ under UV irradiation. Moreover, the reaction of MABr and PbBr_2_ facilitated the conversion of Pb^2+^ to MAPbBr_3_, enabling the extraction of Pb^2+^ from waste products. Figure 24 shows the detailed experimental process and results of this study [60].

When MABr encounters Pb (II) bound to sulfhydryl groups, it leads to the growth of high-fluorescence (MAPbBr_3_) perovskite in situ in sulfhydryl-functionalized mesoporous alumina films. This sulfhydryl modification significantly enhances the extraction ability of mesoporous alumina membranes for Pb (II), as reported by Wang et al. Figure 25 illustrates the selection of a commercial meso-Al_2_O_3_ film with a pore size of 20 nm as a prototype for 3-mercaptopropyltriethoxysilane (MPTS) modification, due to the strong Pb-S bonding. The hydroxyl groups on the surface of the meso-Al_2_O_3_ film facilitate the sulfydryl modification through silane hydrolysis, as shown in Figure 25c. The meso-Al_2_O_3_-SH film efficiently enriches Pb(II) in aqueous solutions, which is subsequently dried in an oven. Excess MABr solution is added to the film, and the solvent is evaporated in an oven at 60 °C. Upon drying, MAPbBr_3_ nanocrystals form on the skeleton or in the holes of the meso-Al_2_O_3_-SH film, exhibiting green fluorescence emission under the excitation of 365 nm ultraviolet light (Figure 25a). The PL intensity of the MAPbBr_3_ nanocrystals increases linearly with the increase of Pb(II) concentration in the sample solution, enabling the visual determination of Pb(II) (Figure 25b). Under optimal conditions, the adsorption capacity of Pb (II) by sulfhydryl-functionalized membranes reaches 94.9 μg/g. Due to the strong extraction ability of Pb (II), the fluorescence opening determination of Pb (II) is realized by forming MAPbBr_3_ perovskite at the site of sulfhydryl-functional alumina film. The LOD is as low as 5 × 10^−3^ μg/mL. This method integrates the extraction and determination of Pb (II) without background, effectively addressing the problem of low concentration detection and promoting the entire analysis process [61].

##### Anion

Chlorine and iodine residual trace elements during tap water disinfection can pose significant potential harm to human health. Park et al. synthesized CsPbBr_3_ perovskite quantum dots (PQDs) cellulose composites using a thermal injection method, as depicted in Figure 26. The composites showed excellent stability, durability, and photoluminescence (PL) properties under various humidity conditions. Initially, cellulose fibers were stabilized with PbBr_2_ precursors to adsorb ions between fibers (Figure 26a). Subsequently, Cs-oleate was added to the PbBr_2_/cellulose composite at a higher temperature of 200 °C, leading to the nucleation of CsPbBr_3_ PQDs and their immediate physical aggregation with cellulose fibers. The successful synthesis of CsPbBr_3_ PQDs and their integration with cellulose nanofibers were confirmed through green PL under UV field (Figure 26c). A portable sensor for early diagnosis was developed based on this composite, which used the rapid anion exchange strategy between halide anions in cellulose fiber PQDs with a high porous structure to detect iodide and chloride ions in real water samples, with detection values of 2.56 mM and 4.11 mM, respectively. CsPbBr_3_ PQDs/cellulose composites have the advantages of using low-cost materials for one-pot continuous synthesis, high sensitivity, reasonable detection time, simple and portable visual detection, flexibility of practical application, and a hydrophilic sensing platform. The response detection time can be shortened to change the color, indicating the possibility of real-time and on-site detection [62]. 

Table 4 summarizes the PNC-based fluorescent probes developed for ion detection. The probes have demonstrated high sensitivity, rapid response times, and ease of use in detecting Cu^2+^, Hg^2+^, Fe^3+^, Zn^2+^, Ag^+^, Pb^2+^, I^−^, and Cl^−^ anions in environmental samples. These features make PNC-based fluorescent probes promising candidates for visual detection of harmful ions with great potential for practical applications.

### 2.2. The Detection of Food-Related Hazardous Substances

#### 2.2.1. Antibiotic Detection

The overuse of antibiotics can lead to the proliferation of resistant bacteria, which can have harmful effects on human health and cause allergic reactions. Furthermore, the presence of antibiotic residues in food, due to their usage in livestock, is a significant concern. Hence, a simple and highly sensitive technique is required for detecting antibiotics in food samples. Roxithromycin (ROX) is a new generation of macrolide antibiotics that mainly targets Gram-positive bacteria, anaerobic bacteria, chlamydia, and mycoplasma. Han et al. have developed a novel CsPbBr_3_-loaded PHEMA MIP nanogel with high stability against water and oxidation, using multifunctional MIP nanogel synthesized from four 2-(hydroxyethyl)methacrylate (HEMA) derivatives with different functions. First, MIP antioxidant-nanogels were prepared with ROX as a template, and then perovskite nanoparticles were loaded into nanogels via in situ synthesis through the hot-injection method. The developed CsPbBr_3_-loaded MIP antioxidant-nanogels exhibited excellent stability to air/water and enhanced stability to water-based solvents. Finally, these nanogels were successfully applied for the selective and sensitive detection of ROX antibiotics in animal-derived food products, as shown in Figure 27. The limit of selectivity and sensitivity to ROX was 1.7 × 10^−5^ μg/mL (20.6 pM), and the detection results demonstrated good recovery, indicating the excellent performance of the developed MIP antioxidant-nanogel loaded with CsPbBr_3_ [63].

Tetracycline (TC) is a class of broad-spectrum antibiotics with phenanthrene nuclei discovered in the 1940s. This class of antibiotics is widely used to treat infections caused by Gram-positive and negative bacteria, intracellular mycoplasma, chlamydia, and Rickettsia. In addition, tetracycline is often used as a growth promoter for animals in some countries, including the United States.

In Figure 28, Jia et al. developed a perovskite-composited ratiometric fluorescent nanosensor with good water stability for detecting tetracycline. The sensor was inspired by the respective advantages of perovskite and rare earth elements. The team prepared rod-like MAPbBr_3_@PbBr(OH) perovskite using the hydration method, with the PbBr(OH) surface layer providing dual advantages of preventing water from entering PNCs and dispersing PNCs in the matrix, resulting in high stability of MAPbBr_3_@PbBr(OH) in water. Under UV irradiation, the perovskite nanorods emitted stable green fluorescence, which served as the fluorescence internal standard for ratiometric fluorescent nanosensors. The silica shell was used to cover the surface of PQDs, which facilitated the subsequent surface modification of europium compounds. Upon the addition of TC, the green fluorescence intensity of MAPbBr_3_@PbBr(OH)@SiO_2_-Cit-Eu decreased slowly, while the energy transfer from TC to Eu^3+^ led to the gradual enhancement of red fluorescence. The detection system exhibited a multi-color visual transformation from green to red fluorescence, which could be applied to the visual rapid detection of TC in actual liquid foods and have great potential application value. The sensor has a good linear response in the range of 0–25 μM, and the detection limit is 11.15 nM. Qualitative and semi-quantitative visual sensing of tetracycline on the surface of a variety of foods (e.g., eggs, apples, bananas, pears, and oranges) can be achieved with the help of color analysis software on a smartphone [64].

Thuy et al. developed a new fluorescent probe for detecting trace amounts of tetracycline (TC) in food samples by preparing highly stable Cs_4_PbBr_6_/CsPbBr_3_ perovskite nanocrystals (PNCs) protected by perfluorooctane triethoxy-silane fluorocarbons on their surface. The high hydrophobic groups formed a waterproof layer, which rendered the PNCs highly stable in water. The fluorescent probes demonstrated high sensitivity and selectivity for TC detection in food samples under optimized conditions, with a limit of detection (LOD) of 76 nM and a linear range of 0.4–10 μM [65].

In Figure 29, Wang et al. developed a novel CsPbBr_3_@BN fluorescent probe for the highly sensitive detection of TC with an LOD as low as 6.5 μg/L. The green-emitting CsPbBr_3_ quantum dots (QDs) were successfully adhered to BN nanosheets, and upon the addition of TC, the CsPbBr_3_@BN fluorescence was sensitively quenched due to electron transfer between them. The fluorescent probe was successfully applied in the determination of TC in honey and milk samples [66]. Furthermore, Wang et al. synthesized a novel fluorescent probe for TC in highly polar ethanol at room temperature. The silicon layer was easily modified by in situ hydrolysis of 3-amino-propyl triethoxysilane (APTES) on the surface of IPQDs at room temperature, producing a new fluorescent probe without water and initiator that could be stably stored in ethanol. The method had high selectivity and sensitivity to TC in ethanol, and the LOD could even reach 76 nM. The fluorescence quenching mechanism was mainly due to electron transfer between TC and IPQDs. The sensor was successfully applied to the detection of trace TC in practical samples. These studies laid a foundation for improving the stability of PNCs and their development in the field of TC detection [67].

Salari et al. developed a chemiluminescent (CL) probe for the highly selective and sensitive determination of cefazolin (CFZ) antibiotic in food, water, and biological samples. The CL probe consists of CsPbBr_3_ quantum dots (QDs) in an organic phase and Fe(II) and K_2_S_2_O_8_ in an aqueous medium. The synthesis of CsPbBr_3_ QDs from CsBr and PbBr_2_ by a solution-based method at room temperature is illustrated in Figure 30. The designed probe exhibited a linear range of 25–300 nM with a low LOD of 9.6 nM and a recovery rate of 94% to 106% for CFZ, enhancing the CL signal of the probe. This research has the potential to advance the development of CL probes for the detection of antibiotics in various fields [68].

By analyzing the chemical formula of ciprofloxacin hydrochloride (C_17_H_18_FN_3_O_3_·HCl·H_2_O), the concentration of its solution can be determined by measuring the Cl^−^ ionic concentration. CsPbBr_3_ NCs have superior optical properties and high anion exchange capacity, enabling precise and sensitive detection of Cl^−^ ionic concentrations. Shi et al. developed colorimetric test strips based on CsPbBr_(3−x)_Cl_x_ NCs for rapid and convenient detection of ciprofloxacin hydrochloride in food. As depicted in Figure 31, at room temperature, CsPbBr_3_ NCs and C_17_H_18_FN_3_O_3_·HCl·H_2_O undergo an anion exchange reaction where some of the Br^−^ ions are replaced by Cl^−^ ions. Upon exposure to different concentrations of ciprofloxacin hydrochloride, the emitted light of CsPbBr_3_NC changes from 513 nm to 442 nm, and the color of the test strip changes immediately after exposure to different ciprofloxacin solutions [69].

Table 5 summarizes the PNC-based fluorescent probes developed for detecting antibiotics in food samples. Fluorescent probes using CsPbBr_3_ and MAPbBr_3_ NCs were successfully applied to detect antibiotics such as ROX, TC, CFZ, and ciprofloxacin hydrochloride. The results show excellent stability, high selectivity and sensitivity, low LOD, and good recovery rate. Furthermore, the use of color analysis software in smartphones enables qualitative and semi-quantitative visual sensing of antibiotics on various food surfaces, which provides a basis for improving the stability of PQDs and their potential for use in detection applications.

#### 2.2.2. Detection of Microbial Toxins, Pathogens, and Carcinogens

Aflatoxin B_1_ (AFB_1_) contamination is one of the most serious problems in food safety. Su et al. have developed a photoelectrochemical (PEC) immunosensing platform for sensitive detection of AFB_1_ in peanut and corn samples using CsPbBr_3_ NCs and amorphous TiO_2_ [CsPbBr_3/_a-TiO_2_]. AFB_1_-BSA conjugates labeled with alkaline phosphatases (ALP) were used as competitors in the competitive immunoreaction triggered on anti-AFB_1_ antibody-coated microplates. In Figure 32, photoelectrochemically active CsPbBr_3_/a-TiO_2_ nanocomposites were dropwise added onto a fluorine-doped tin oxide (FTO) electrode. The ALP collected on the microplate hydrolyzed ascorbic acid-2-phosphate (AAP) to produce ascorbic acid (AA), which enhanced the photocurrent of CsPbBr_3_/a-TiO_2_ nanocomposites. The concentration of AFB_1_ in food can be monitored by detecting the photocurrent of the CsPbBr_3_/a-TiO_2_-modified electrode. The PEC platform exhibited a working range of 0.01~15 ng/mL and a low detection limit of 2.8 pg/mL. The accuracy of this method was found to be acceptable compared with the reference aflatoxin enzyme-linked immunosorbent assay (ELISA). The developed PEC immunosensing platform shows promise in the sensitive and accurate detection of AFB_1_ in foodstuffs [70]. In addition, Li et al. synthesized CH_3_NH_3_PbBr_3_ (MAPB) QDs@SiO_2_ nanospheres via the encapsulation of MAPB QDs with a network structure of APTES hydrolyzed and condensed with SiO_2_. They constructed an electrochemical luminescence (ECL) platform using MAPB QDs@SiO_2_ for the highly selective and ultra-sensitive detection of AFB_1_, with a LOD of 8.5 fg/mL. The sensor exhibited good recovery rates for AFB_1_ in real corn oil samples [71]. 

Staphylococcus aureus contamination of food at 20–37 °C can result in the production of enterotoxin, causing acute gastroenteritis symptoms such as nausea, vomiting, and diarrhea within 1–5 h of consumption. Food poisoning is mainly caused by enterotoxins rather than bacteria. Staphylococcal enterotoxin (SEs) consists mainly of S. aureus enterotoxins A, B, C, D, and E (SEA, SEB, SEC, SED, and SEE), and the minimum dose of enterotoxin required to cause food poisoning is 1–7.2 g/kg body weight. Xu et al. developed CsPbBr_3_@mesoporous silica nanomaterials (MSN) and converted it into CsPb_2_Br_5_@MSN in an aqueous phase for immunoassay of Staphylococcal enterotoxin (SEs). This additive-free surface-enhanced Raman scattering (SERS) immunoassay can be used for the simple, sensitive, and reproducible detection of SEC. The research can be extended to the development of various perovskite composites, providing the possibility of exploring more SERS detection probes for food safety monitoring [72].

In food processing, waxing can maintain the fresh flavor of fruit. However, artificial wax applied to the surface of fruit often contains morpholine, which is easily converted into nitroso-morpholine in the human body and can cause liver or kidney cancer. In addition, artificial wax also contains protein substances that can cause allergic reactions. Ye et al. [73] prepared a lead-free Cs_2_PdBr_6_ perovskite humidity sensor, which was used for the first time to detect artificial wax on fruit. The manufactured humidity sensor has a response time of 0.7 s and a recovery time of 1.7 s. The impedance varies by 10^5^ Ω when the relative humidity increases from 11% to 95%. The oleic acid-modified Cs_2_PdBr_6_ sensor has the advantages of high sensitivity and fast response/recovery. It can be used to distinguish the waxy/wax-free apples and oranges from the surface moisture.

Table 6 summarizes PNC-based fluorescent probes for the detection of microbial toxins, pathogens, and carcinogens in food. PNCs of sPbBr_3_, CH_3_NH_3_PbBr_3_, and lead-free Cs_2_PdBr_6_ were used to detect B_1_ (AFB_1_), SEs, nitrite, and artificial wax on fruit and showed high sensitivity and selectivity and has a wide range of application potential. 

#### 2.2.3. Detection of Food Spoilage Gas

The odor of ammonia is a result of protein breakdown by microorganisms in spoiled food. To detect ammonia, Huang et al. developed a highly sensitive and selective CsPbBr_3_ QDs film sensor that is fully reversible. Real-time dynamic passivation of PQDs was studied by monitoring changes in PL intensity, as shown in Figure 33. The sensing device consists of a silicon detector and a PQDs film, excited at 365 nm under the flow of ammonia gas. Exposure to ammonia gas led to a remarkable increase in fluorescence of the QDs film, which could even be observed by the naked eye, and returned to its original state upon removal of the gas. The response–recovery cycles of this process are shown in the schematic diagram in Figure 33. The sensor can detect ammonia in a wide range of 25–350 ppm, with an LOD as low as 8.85 ppm, and a fast response time of ≈10 s and a recovery time of ≈30 s are achieved, respectively. This novel sensor has potential applications in food safety and environmental monitoring [74].

Hydrogen sulfide (H_2_S) is a toxic gas that emits a rotten egg smell, and exposure to high levels of H_2_S in food can be detrimental to human health. Li et al. developed water-soluble inorganic PQDs (CsPbBr_3_@PEG-PCL nanoparticles) by encapsulating CsPbBr_3_ QDs with a block copolymer of polyethylene glycol polycaprolactone (PEG-PCL), as illustrated in Figure 34a. The fluorescence of CsPbBr_3_@PEG-PCL nanoparticles remains stable in aqueous solution for at least 15 days under light and room temperature conditions, making it ideal for H_2_S detection. The fluorescence intensity of CsPbBr_3_@PEG-PCL nanoparticles is quenched by H_2_S, and thus, a highly sensitive H_2_S fluorescent probe can be constructed based on fluorescence intensity quenching. The fluorescence signal has a linear relationship with H_2_S concentration in the range of 0–32.00 μM, with a limit of detection (LOD) of 37.65 nM. CsPbBr_3_-based sensors are also capable of accurately detecting H_2_S in wine samples, cells, and zebrafish [75]. Another type of water-soluble inorganic PQDs (CsPbBr_3_@SBE-β-CD nanoparticles) was obtained by passivating the surface ligands with sulfobutyl ether-β-cyclodextrin (SBE-β-CD). This nanoparticle can be used as a photothermal probe to react with H_2_S and produce a substance with high photothermal effect. Under 808 nm laser irradiation, the temperature change of the system was measured using a thermometer as a signal reading device to quantitatively analyze H_2_S, as shown in Figure 34b. There is a linear relationship between temperature and H_2_S concentration in the range of 0.5–6000.0 μM, with a LOD of 0.19 μM. This sensor can achieve the quantitative detection of H_2_S in water medium [76].

Zu et al. proposed a synthesis method for all-inorganic CsPbBr_3_ QDs that can be directly stored in aqueous solution, maintaining excellent fluorescence stability and dispersion, as illustrated in Figure 35. By adding CTAB aqueous solution, CsPbBr_3_ QDs and mineral oil can be rapidly processed by ultrasonic to form CsPbBr_3_@CO complex. In aqueous solution, when the concentration of CTAB as a surfactant exceeds the critical micellar concentration (CMC), spherical or rodlike micelles may be formed, with hydrophobic nuclei surrounded by hydration shells. These micelles can encapsulate hydrophobic substances, enhancing their solubility and dispersion in water. As fluorescent probes, these micelles exhibit good selectivity for the detection of hydrogen sulfide (H_2_S) in food and have a short detection time. The fluorescent probe demonstrates a linear relationship between the H_2_S concentration and the fluorescence intensity in the range of 0.15–105.0 μM, with an LOD of 53.0 nM, satisfying the H_2_S detection requirements for most food or biological samples [77].

Table 7 summarizes fluorescent probes based on PNCs for detecting food spoilage. Modified CsPbBr_3_ QDs exhibit high sensitivity and selectivity in detecting gases such as ammonia and H_2_S in wine, beer, and biological samples (zebrafish), with rapid response and recovery times.

#### 2.2.4. Detection of Harmful Additives in Food

Sudan dyes, classified as carcinogens, are commonly used as phenylazo derivatives in industry but are strictly prohibited from use in food. He et al. utilized CsPbX_3_ PQDs of varying colors within molecularly imprinted polymers as fluorescent probes for the sensitive and selective detection of Sudan Red I in food. The fluorescence intensity of MIPs-CsPbX_3_ microspheres was significantly quenched after loading Sudan Red I. The linear response was good within the range of 0.5–150 μg/L, with an LOD of 0.3 μg/L and recoveries between 95.27–105.96% [2]. Wu et al. developed a simple and effective fluorescence detection platform for Sudan I-IV utilizing CsPbBr_3_ PQDs, as shown in Figure 36. Under UV excitation, the CsPbBr_3_ QDs exhibited bright green fluorescence, which was efficiently quenched by Sudan I-IV due to significant overlap between the absorption band of Sudan I-IV and the fluorescence spectrum of CsPbBr_3_ QDs. After optimization, the quantification of fluorescence quenching efficiency of CsPbBr_3_ QDs was found to be correlated with the logarithmic concentrations of Sudan I-IV (100–10000, 0.1–1000, 0.1–2000, and 0.4–1000 ng). The LODs were 3.33, 0.03, 0.03, and 0.04 ng/mL, respectively [78]. 

Rhodamine 6G (Rh6G) is an anthraquinone dye that has found wide application in the textile and food industries. However, due to its teratogenic, carcinogenic, and other biological toxicities to both humans and animals, it is crucial to detect it effectively in low concentration ranges. Chan et al. synthesized monodispersed PEGylated CsPbBr_3_/SiO_2_ QDs using the LARP synthesis method at an ambient temperature of 23 °C. In Figure 37, the CsPbBr_3_/SiO_2_ QDs were encapsulated in mPEG-DSPE phospholipid micelles to provide excellent aqueous solubility and stability. This nanosensor exhibits good sensitivity and selectivity for Rh6G and can also detect it in biological samples containing complex interference backgrounds. The operation range is 0–10 mg/mL, and the LOD is 0.01 mg/mL [79]. Similarly, Wang et al. prepared a CsPbBr_3_ PQDs/polystyrene fiber film (CPB QDs/PS FM) using a one-step electrospinning method. Due to the excellent optical properties of CPB QDs, the ultra-low LOD for Rh6G detection is 0.01 ppm, and FRET efficiency is 18.80% in 1 ppm Rh6G aqueous solution [80].

Furthermore, RhoB is a harmful substance that often appears in printing and dye wastewater or as an additive in food. Traditional detection systems are not sensitive enough and often require a large number of sample solutions (>1 mL) to concentrate to a higher concentration. Ni et al. fabricated a reusable perovskite nanocomposite fiber paper using microwave and electrospinning methods, which consisted of CsPbBr_3_ QDs grown in situ in a high concentration solid polymer fiber. This perovskite fiber paper can be easily recycled and has the advantages of ultra-sensitive detection (0.01 ppm), rapid detection (<3 min), and minimal dose (<25 μL), which are obviously superior to traditional detection systems [81].

Shi et al. synthesized oil-soluble all-inorganic CsPbBrI_2_ QDs using the thermal injection method for the detection of basic yellow dyes that are illegally added to food and beverages. Their results showed a linear range of 1–500 μg/mL, a LOD of 0.78 μg/mL, and recoveries of 95.27–98.84% at levels of 8, 40, 80, and 400 μg/mL, with relative standard deviations of less than 3.52% [82].

Li et al. developed an ultra-sensitive fluorescent nanosensor for the detection of melamine using PNCs (CsPbBr_3_NCs@BaSO_4_). Figure 38 illustrates the nanosensor’s principle. The negatively charged citrate-stabilized AuNPs were mixed with positively charged CsPbBr_3_ NCs@BaSO_4_, causing them to assemble together through electrostatic interaction. This aggregation led to strong fluorescence quenching of the CsPbBr_3_ NCs@BaSO_4_ due to the inner filter effect of the AuNPs. However, upon the addition of melamine, the color of the solution changed from wine red to blue–grey, indicating the agglomeration of the AuNPs. This aggregation decreased the inner filter effect of the AuNPs, resulting in a recovery of the fluorescence of the CsPbBr_3_ NCs@BaSO_4_. The nanosensor was used for the determination of trace melamine in dairy products, achieving a LOD of 0.42 nmol/L and a relative standard deviation of 4.0% by repeated determination of 500.0 nmol/L (*n* = 11) [83].

To achieve efficient electrochemical luminescence (ECL) in PQDs, Sun et al. proposed the synthesis of Cs_4_PbBr_6_@CsPbBr_3_ PQD nanoacanthospheres (PNAs) with optimized conditions resulting in ECL activity four times higher than 3D (CsPbBr_3_) PQDs, which was then applied to Bisphenol A (BPA) ECL sensing [84].

Table 8 summarizes the use of modified CsPbBr_3_ and CsPbBrI_2_ QDs as PNC-based fluorescent probes for detecting harmful additives in food. These probes showed good selectivity and sensitivity in detecting Sudan I–IV, Rh6G, RhoB, basic yellow dye, melamine, and bisphenol A, with the advantages of rapid determination and minimal dose. This demonstrates their potential practical applications and superiority over traditional detection systems.

#### 2.2.5. Edible oil Quality Inspection

Edible oil undergoes complex changes during storage, with the most important being the decomposition of oil into free fatty acids and other products under the influence of light, heat, air, and other factors, commonly referred to as rancidity. Peroxide is an intermediate product formed during the process of oil oxidation and rancidity, which is unstable and can decompose into aldehydes, ketones, and other oxides. This not only affects the flavor and nutritional value of food but also poses risks to human health. China’s National Testing Standard GB/T 2716-2018 specifies that the peroxide value of edible oil should be <0.25 g/100 g. Currently, the peroxide value of cooking oil is usually determined by titration or potential methods, which are not reproducible, sensitive, and environmentally friendly, and the methods applied are cumbersome to operate. Although spectrophotometry and high-performance liquid chromatography have been developed to overcome these shortcomings, there is still a need for real-time and visual detection methods that are simple and rapid.

The peroxide number of edible oil is related to its quality, and classical determination methods for the peroxide number are unsatisfactory due to their complexity and poor reproducibility in the analytical process and their incapability of field rapid detection. To overcome this, a colorimetric sensing method based on the wavelength shift of CsPbBr_3_ NCs with the addition of oleylammonium iodide (OLAM-I) was developed for the determination of peroxide number in edible oil. As shown in Figure 39, by performing halide exchange with OLAM-I, the fluorescence emission wavelength of CsPbBr_3_ NCs gradually red-shifts, with the degree of red-shift proportional to the amount of OLAM-I added. This results in a color change from green to yellow and finally to red, reflecting the halogen-exchange characteristics of CsPbBr_3_ NCs and the redox reaction between OLAM-I and peroxides in edible oil. This method enables the visual detection of the peroxidation value of edible oil samples, with a detection process that only takes about 15 min and has been proven to be convenient and accurate [85]. 

The quality of edible oil has a significant impact on human health, with excessive acid number (AN), 3-chloro-1,2-propylene glycol (3-MCPD), and moisture content (MC) being the key factors to monitor. To address this issue, Zhao et al. developed orange luminescent oil-soluble CsPbBr_1.5_I_1.5_ QDs and applied them to detect AN, 3-MCPD, and MC in edible oil by employing fluorescence quenching and wavelength shift. The mechanism diagram of the detection method for hazardous substances in edible oil is shown in Figure 40. By adding varying concentrations of benzoic acid into the oil, the orange-emitting fluorescence of oil-soluble CsPbBr_1.5_I_1.5_ QDs was quenched by edible oil with different ANs, and a good linear relationship was observed between the fluorescence intensity of CsPbBr_1.5_I_1.5_ QDs and AN concentration, enabling quantitative detection of AN in edible oil. Additionally, the unique halogen exchange reaction between CsPbBr_1.5_I_1.5_ QDs and 3-MCPD allowed for the sensitive and specific detection of 3-MCPD, as the emission peak of CsPbBr_1.5_I_1.5_ QDs blue-shifted with increasing 3-MCPD concentration. To monitor the MC of edible oil, water-sensitive PQDs were prepared to establish ratiometric fluorescent probes. The orange-emitting CsPbBr_1.5_I_1.5_@MSNs were used as detection probes, which were generally quenched when exposed to water, while green-emitting quasi-two-dimensional (2D) CsPbBr_3_ NSs served as reference probes due to their stable fluorescence properties in aqueous solution. The ratiometric fluorescent probes based on these two materials were established for MC detection in edible oil, with limits of detection (LODs) of 0.71 mg KOH/g, 39.8 μg/mL 3-MCPD, and 0.45% MC. By leveraging this detection principle, the researchers achieved quantitative detection of AN, 3-MCPD, and MC in edible oil, demonstrating great potential for monitoring the quality of edible oil [86]. 

Accurately detecting trace moisture in edible oil is crucial to ensure its quality and safety. Although all inorganic PQDs are potential candidates for optoelectronic applications, their stability is often restricted by environmental factors, especially high polarity materials. In contrast, perovskite-structured QDs are highly suitable as polarity sensors due to their instability to polar materials. To address this, CsPbBr_3_ PQDs were synthesized at low temperature, which were modified with a functional ligand dimethyl aminoterephthalate (CsPbBr_3_@DMT-NH_2_ QDs). The resulting dual-emission QD displayed highly sensitive fluorescence turn-on/off dual response and a distinct fluorescence color change from green to blue in the presence of water. This enabled the establishment of ratiometric fluorescence sensing and visual ratiometric chromaticity detection methods for water assay, with ultralow LODs of 0.006% (*v*/*v*) and 0.01% (*v*/*v*), respectively. These methods accurately detected trace water in edible oils (with a recovery rate of 93.0% to 108.0%, and a relative error of ≤5.33%). The study also revealed that the high polarity levels of protic solvents can disintegrate the QDs, while the relatively low polarity levels promote their aggregation, ultimately leading to green fluorescence quenching. Furthermore, the deprotonation and polarity of water enhance the blue fluorescence of DMT-NH_2_ by promoting excited-state intramolecular proton transfer and intramolecular charge transfer, as shown in Figure 41. This is the first report on dual-response ratiometric fluorescence PQDs for ultrasensitive water detection in edible oils [87].

To ensure the quality and safety of edible oil, the analysis of total polar substance (TPM) is crucial. However, the development of appropriate TPM analysis methods has been challenging in the food safety field. A paper-based fluorescent probe using CsPbBr_3_ QDs was synthesized and utilized for rapid detection of TPM in different types of edible oils. The results showed a linear relationship for olive oil (17–31.5%), soybean oil (25–31.5%), and sunflower oil (21.5–33%). Real-time monitoring of TPM content was achieved using the probe, and the potential for a more rapid detection method was explored. These findings indicate that CsPbBr_3_ QDs can serve as a new fluorescent probe for the rapid detection of TPM in edible oil, with practical application value. Furthermore, Table 9 summarizes the use of PNC-based fluorescent probes for edible oil quality inspection, including modified CsPbBr_3_ NCs and CsPbBr_1.5_I_1.5_ QDs, which demonstrate the convenience and accuracy of detecting peroxide value, AN, 3-MCPD, MC, and TPM. These perovskite nanomaterials not only expand the application of bioanalysis but also provide a new material and method for monitoring the safety of oil-phase foods in multiple ways [88].

## 3. Summary and Prospects

In summary, the fluorescent probe based on PNCs is expected to replace the traditional inorganic QDs because of its high fluorescence quantum yield and sensitivity based on PNCs to detect hazardous substances in agriculture and food. Nowadays, PNCs have been used in agro-ecological environment to detect pesticide residue, environmental pollutant, and harmful ions. At the same time, PNCs have also been applied in food to detect antibiotics, microorganisms, food spoilage gases, additives, and edible oil quality. Compared with the fluorescence analysis method constructed by traditional QDs, PNCs as fluorescent probes are just in their infancy. Therefore, there are still many problems in the practical application of PNCs, and more research is needed.
(1)The external environment (light, humidity, temperature, etc.) is easy to affect photoelectric properties of PNCs. Although, many modification methods have been proved to be effective to achieve PNCs in aqueous solutions, some of these methods are complex or cannot be used as fluorescent probes in the field of agriculture and food. Simple and novel water-stabilized method that is suitable for detection in agriculture and food should be further developed in the future. For example, the exploration of organic semiconductor ligands to prepare PNCs materials with good optoelectrical properties and heterostructures.(2)Lead is an important component of PNCs structure, and it is a huge obstacle, which hinders PNCs in practice in agriculture and food. Non-contact detection (gas or separated liquid samples) may be more suitable for the lead-based PNCs fluorescent probe. The development of novel and efficient lead-free PNCs fluorescent probe is more preferred and should be the future trend in this area.(3)The specificity of the fluorescent probe based on PNCs can be further developed to selectively detect specific analyte. Combined with molecular imprinting techniques, it has been proved to be an effective way to improve its selectivity. Especially, the novel characterizing methods should be paid more attention, such as MIECL platform, PEC immunosensing platform, etc.(4)The integration of fluorescent probe based on PNCs, excitation light source, and fluorescence detector into a portable instrument is still a big challenge. The future practice of fluorescent probe based on PNCs in the field of agriculture and food should tend to portable, real-time, and visualized detection. Fluorescent probe based on PNCs prefer to be developed into the form of test strip. Ratio fluorescence is a good method to get visual sensing with high accuracy. In addition, with the help of the color analysis software in smart phones, qualitative and semi-quantitative visual sensing can be easily realized, which has a good application prospect.

In short, with the deepening of research, fluorescent probe based on PNCs should be constantly improved and innovated. The development of highly sensitive, highly selective, portable, and smart fluorescent probe based on PNCs in the field of agriculture and food is expected.

## Figures and Tables

**Figure 1 polymers-15-02873-f001:**
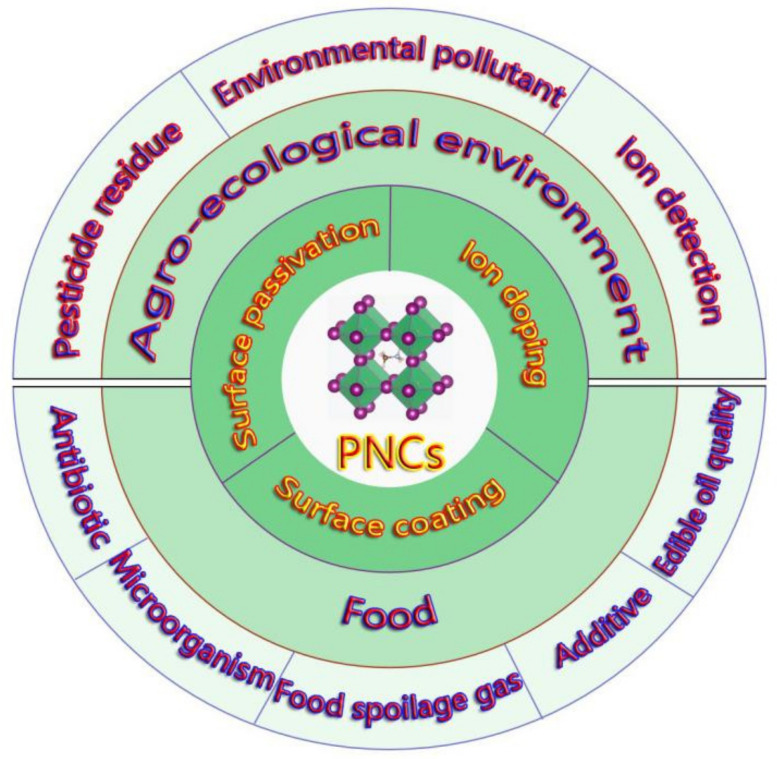
Schematic illustration of the outline.

**Figure 2 polymers-15-02873-f002:**
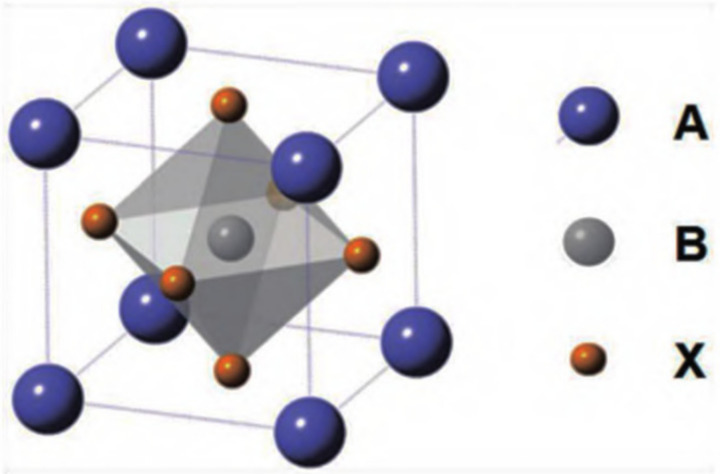
Typical ABX_3_ perovskite crystal structure. Reproduced from ref. [7] Copyright (2019), with permission from Royal Society of Chemistry.

**Figure 3 polymers-15-02873-f003:**
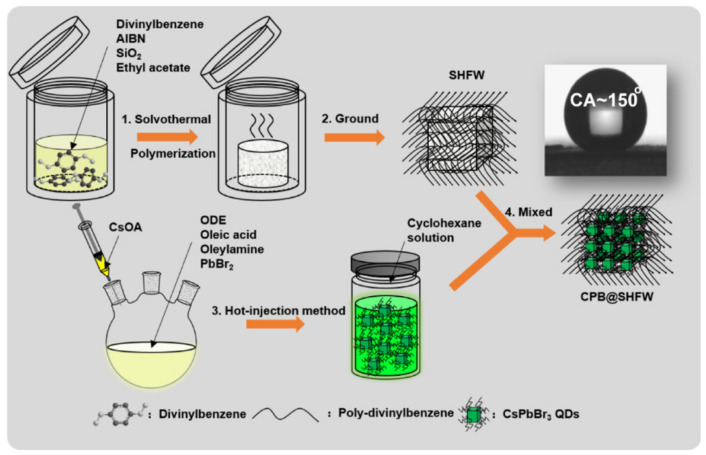
Schematic illustration for preparation of CPB@SHFW composites. Reproduced from ref. [19] Copyright (2019), with permission from American Chemical Society.

**Figure 4 polymers-15-02873-f004:**
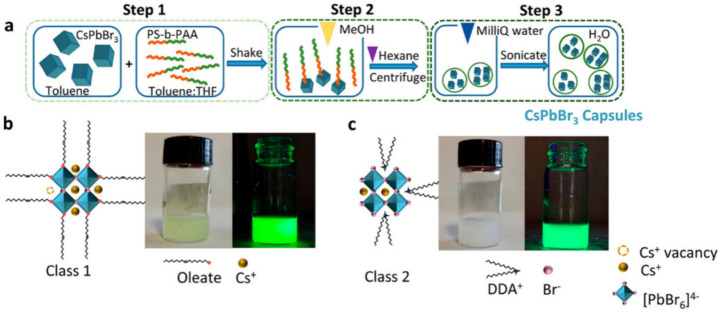
(**a**) Schematic illustration of the capsules embedding the CsPbBr_3_ NCs. (**b**,**c**) Photographs of aged samples of (**b**) Cs-oleate and (**c**) DDAB-coated CsPbBr_3_ NCs dispersed in water under normal and UV light. Reproduced from ref. [20] Copyright (2022), with permission from American Chemical Society.

**Figure 5 polymers-15-02873-f005:**
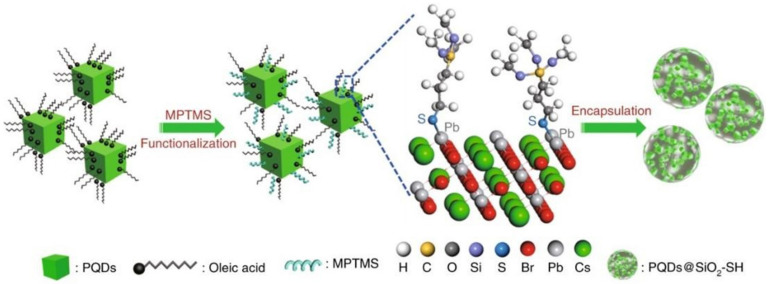
Synthesis of wr-PNDs by encapsulating into SiO_2_. Reproduced from ref. [21] Copyright (2020), with permission from Springer Nature.

**Figure 6 polymers-15-02873-f006:**
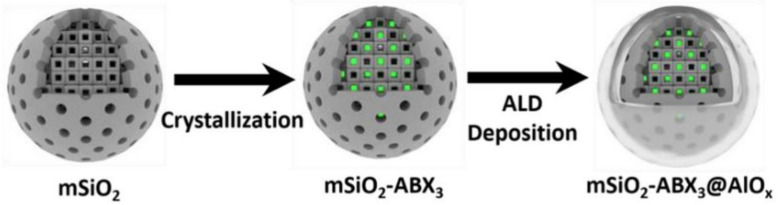
Synthetic procedure of the mSiO_2_-ABX_3_@AlO_x_ (A = MA, FA, Cs, B = Pb, Mn, X = Cl, Br, I) NCs. Reproduced from ref. [22] Copyright (2020), with permission from Elsevier.

**Figure 7 polymers-15-02873-f007:**
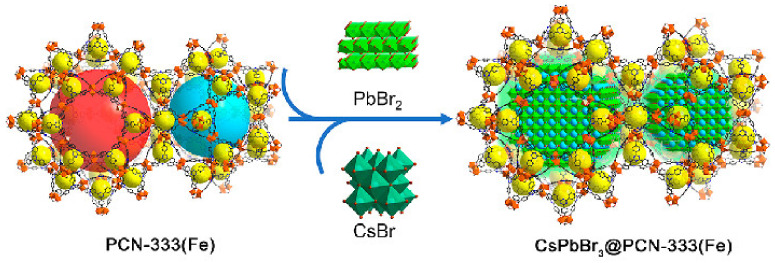
Schematic illustration of the preparation of CsPbBr_3_@PCN-333(Fe). Reproduced from ref. [24] Copyright (2021), with permission from American Chemical Society.

**Figure 8 polymers-15-02873-f008:**
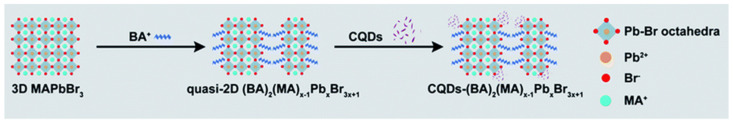
Schematic representation of metal halide perovskites with 3D and quasi-2D structures. Reproduced from ref. [32] Copyright (2021), with permission from Royal Society of Chemistry.

**Figure 9 polymers-15-02873-f009:**
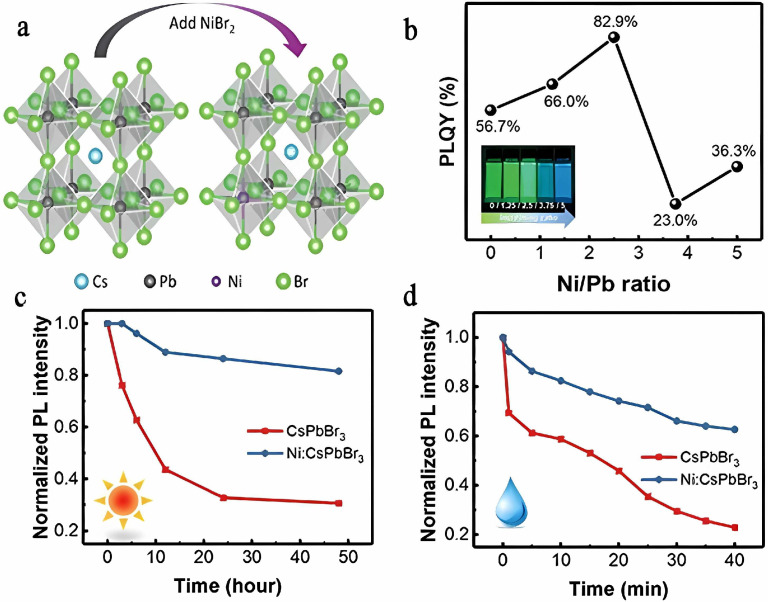
(**a**) Schematic of CsPbBr_3_ PNCs with and without Ni^2+^ substitution. (**b**) PLQY with Ni/Pb feed molar ratios of 0, 1.25, 2.5, 3.75, and 5; insets show photographs of Ni:CsPbBr_3_ PNCs in hexane under 365 nm UV irradiation. Normalized PL intensity of CsPbBr_3_ and Ni:CsPbBr_3_ PNCs (Ni/Pb = 2.5), (**c**) under constant UV radiation (365 nm, 4 W), and (**d**) in deionized water. Reproduced from ref. [37] Copyright (2021), with permission from Wiley-VCH.

**Figure 10 polymers-15-02873-f010:**
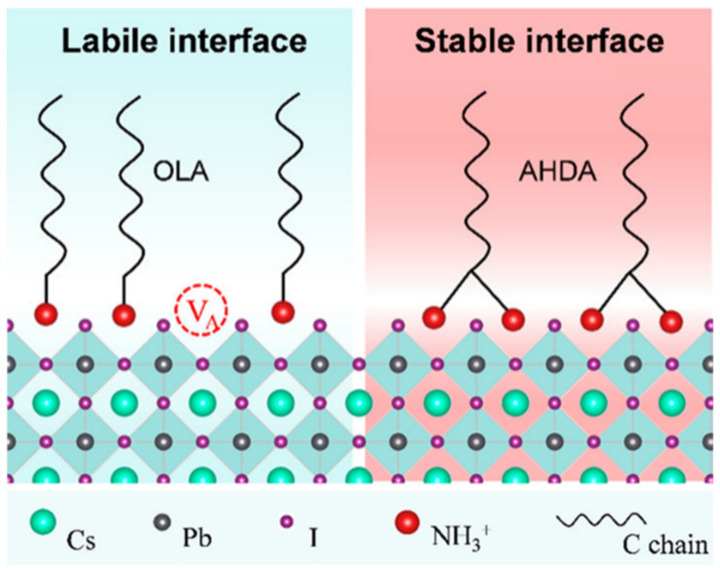
Multi-amine AHDA ligands stabilize the CsPbI_3_ surface. Reproduced from ref. [38] Copyright (2022), with permission from American Chemical Society.

**Figure 11 polymers-15-02873-f011:**
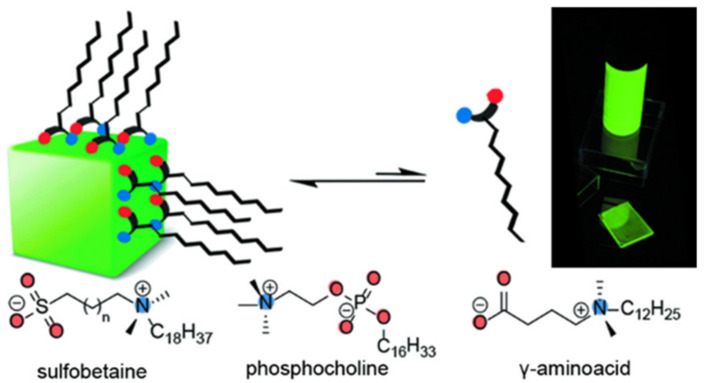
Surface engineering of PNCs with zwitterionic organic ligands, including 3-(N,N-dimethyloctadecylammonio) propanesulfonate (*n* = 1), N-hexadecylphosphocholine, and N,N-dimethyldodecylammoniumbutyrate. Reproduced from ref. [39] Copyright (2019), with permission from Wiley-VCH.

**Figure 12 polymers-15-02873-f012:**
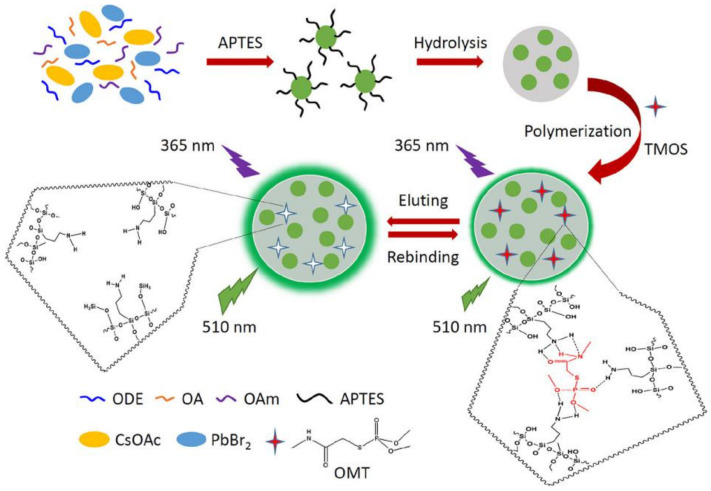
Schematic illustration of the preparation of the MIPs@CsPbBr_3_ QD sensor. Reproduced from ref. [4] Copyright (2018), with permission from American Chemical Society.

**Figure 13 polymers-15-02873-f013:**
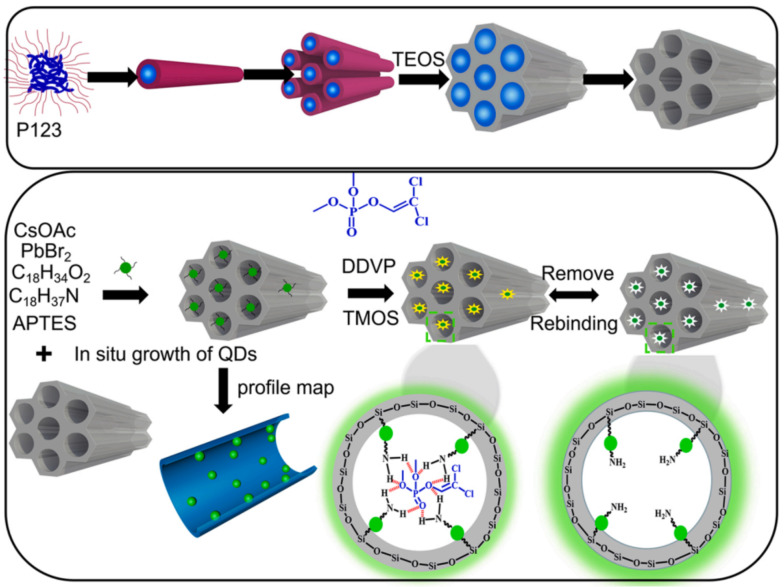
Schematic illustration of the preparation of QDs-encapsulated MIMS. Reproduced from ref. [46] Copyright (2020), with permission from Elsevier.

**Figure 14 polymers-15-02873-f014:**
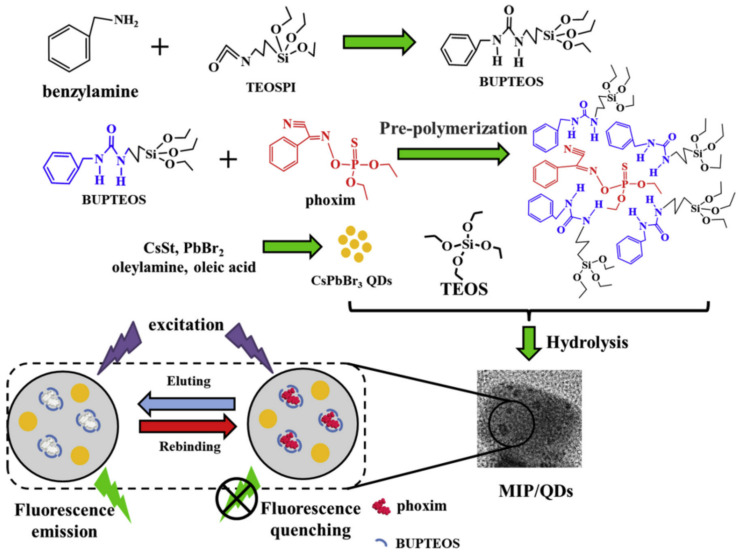
Schematic illustration of the preparation of MIP/CsPbBr_3_ QDs composites. Reproduced from ref. [47] Copyright (2019), with permission from Elsevier.

**Figure 15 polymers-15-02873-f015:**
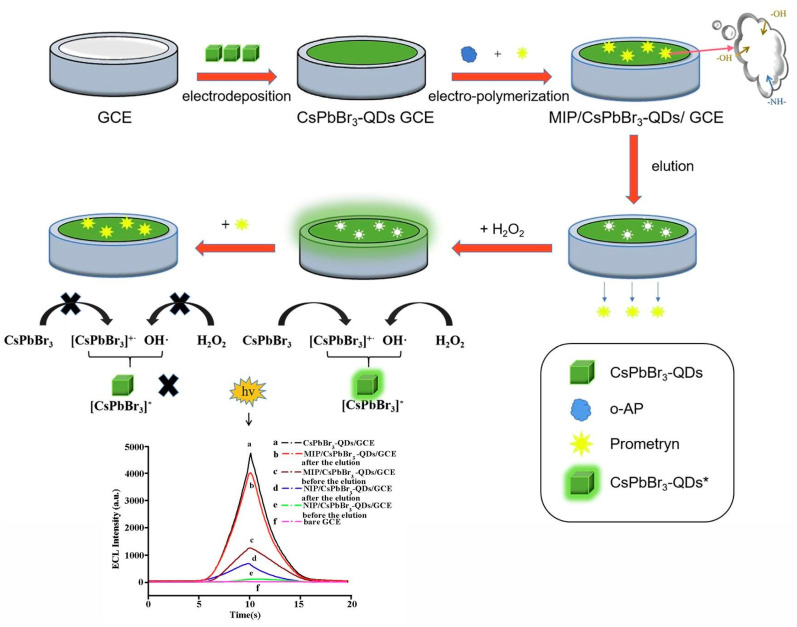
Schematic illustration for the fabrication process and prometryn determination principle of ultrasensitive MIECL sensor. Reproduced from ref. [49] Copyright (2022), with permission from Elsevier.

**Figure 16 polymers-15-02873-f016:**
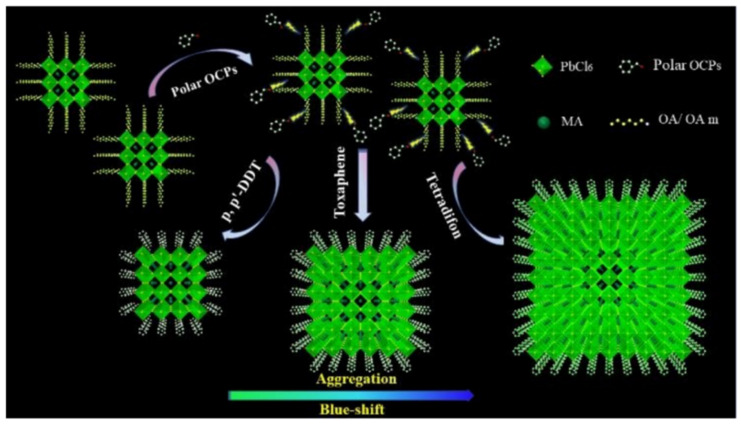
Schematic illustration of the reaction mechanism for MAPB-QDs and polar OCPs. Reproduced from ref. [51] Copyright (2020), with permission from Royal Society of Chemistry.

**Figure 17 polymers-15-02873-f017:**
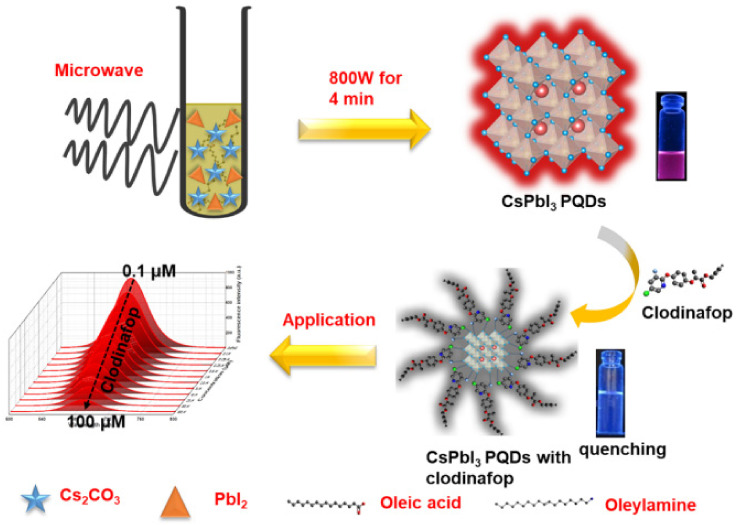
Schematic illustration of microwave synthesis of CsPbI_3_ PQDs for sensing of clodinafop. Reproduced from ref. [1] Copyright (2022), with permission from American Chemical Society.

**Figure 18 polymers-15-02873-f018:**
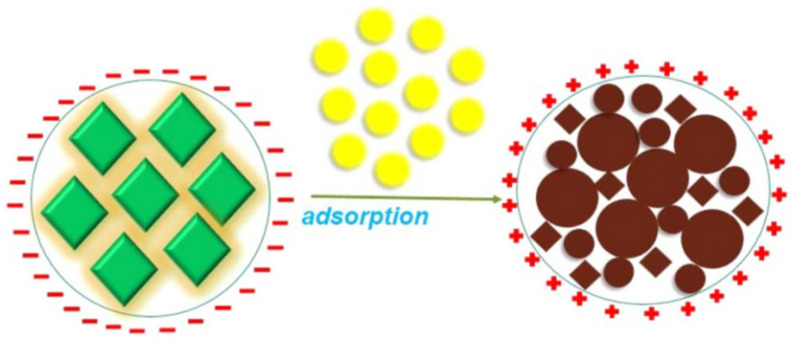
Adsorption mechanism UO_2_^2+^ ions on CsPbBr_3_ PQDs due to the zeta potential process. Reproduced from ref. [53] Copyright (2020), with permission from Elsevier.

**Figure 19 polymers-15-02873-f019:**
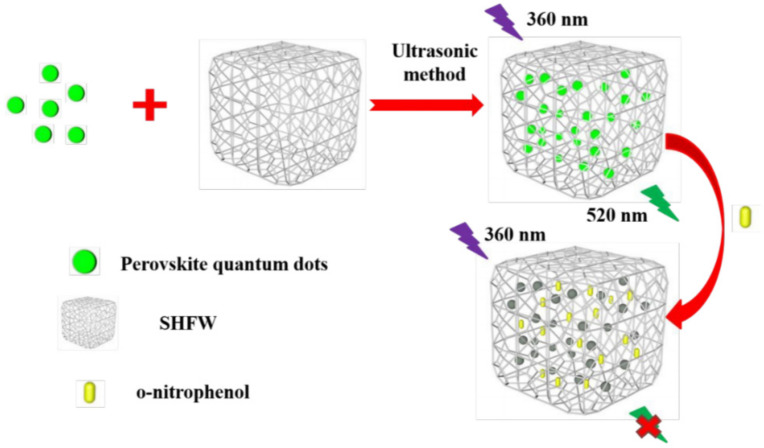
The schematic diagram of the synthetic CsPbBr_3_@SHFW for fluorescence detection of ONP. Reproduced from ref. [54] Copyright (2021), with permission from Royal Society of Chemistry.

**Figure 20 polymers-15-02873-f020:**
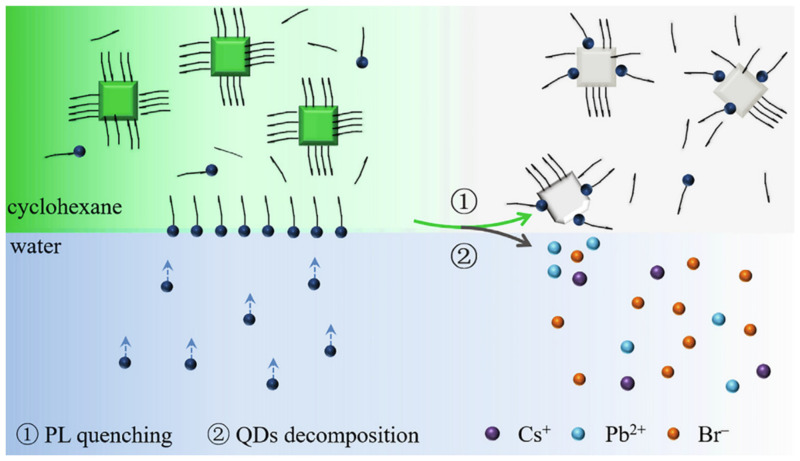
Schematic diagram for phase transfer of Cu^2+^ from water to cyclohexane induced PL quenching (①) and decomposition (②) of CPB QDs. Reproduced from ref. [56] Copyright (2021), with permission from Elsevier.

**Figure 21 polymers-15-02873-f021:**
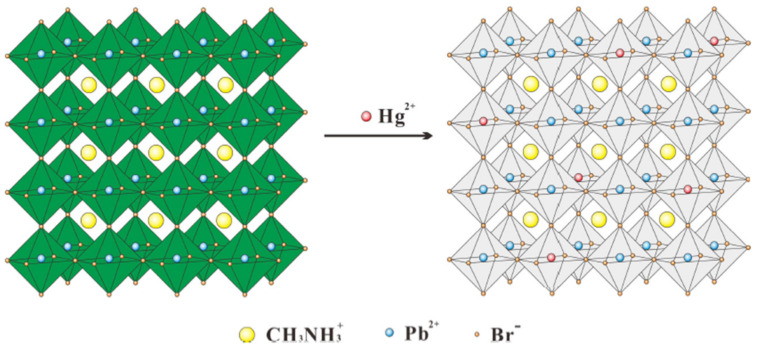
An illustration of the ion exchange process on the surface of CH_3_NH_3_PbBr_3_ QDs. Reproduced from ref. [57] Copyright (2017), with permission from Elsevier.

**Figure 22 polymers-15-02873-f022:**
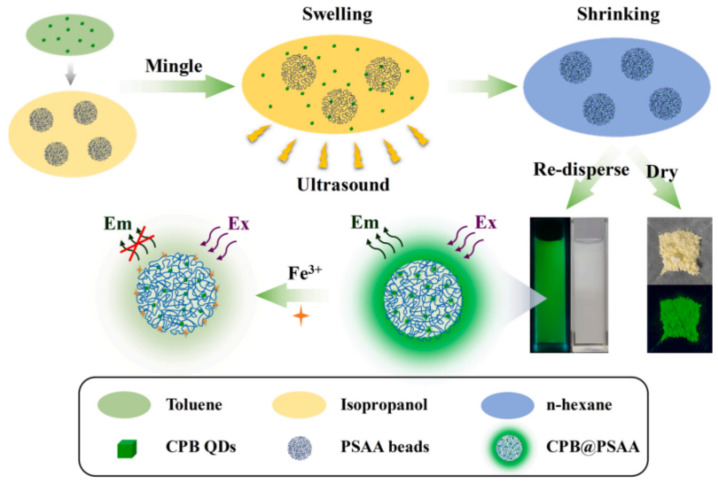
Synthetic diagram of the preparation of CPB@PSAA composites for the detection of iron ion. Reproduced from ref. [58] Copyright (2020), with permission from Elsevier.

**Figure 23 polymers-15-02873-f023:**
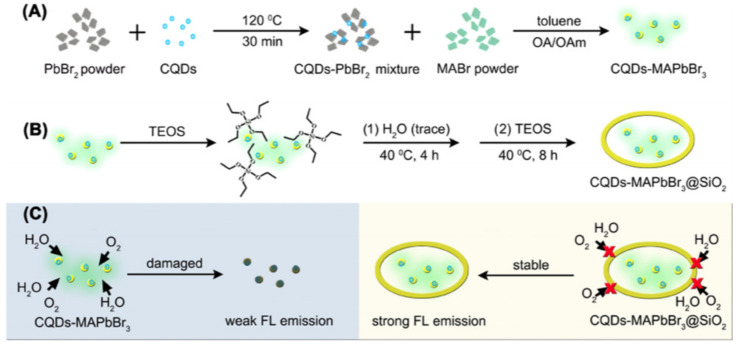
Schematic illustration of the synthesis process of (**A**) CQDs-MAPbBr_3_, (**B**) CQDs-MAPbBr_3_@SiO_2_, and (**C**) Comparison of CQDs-MAPbBr_3_ and CQDs-MAPbBr_3_@SiO_2_ under air and moisture conditions. Reproduced from ref. [59] Copyright (2019), with permission from American Chemical Society.

**Figure 24 polymers-15-02873-f024:**
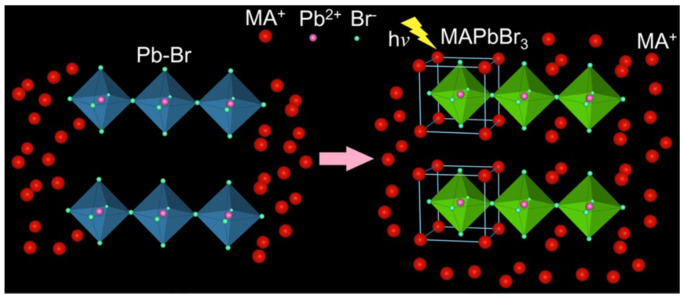
Schematic illustration for the luminescent response of the MABr solution to Pb^2+^. Reproduced from ref. [60] Copyright (2019), with permission from Springer Nature.

**Figure 25 polymers-15-02873-f025:**
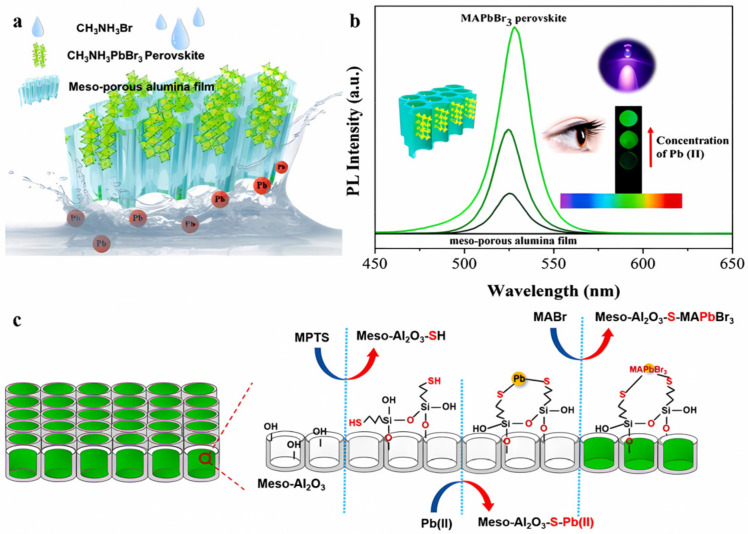
(**a**) Schematic of Pb(II) extraction and MAPbBr_3_ perovskite formation. (**b**) Schematic of fluorescence turn-on and Pb(II) determination. (**c**) Schematic of sulfydryl functionalization, Pb(II) enrichment, and on-site conversion to MAPbBr_3_ perovskite. Reproduced from ref. [61] Copyright (2021), with permission from Elsevier.

**Figure 26 polymers-15-02873-f026:**
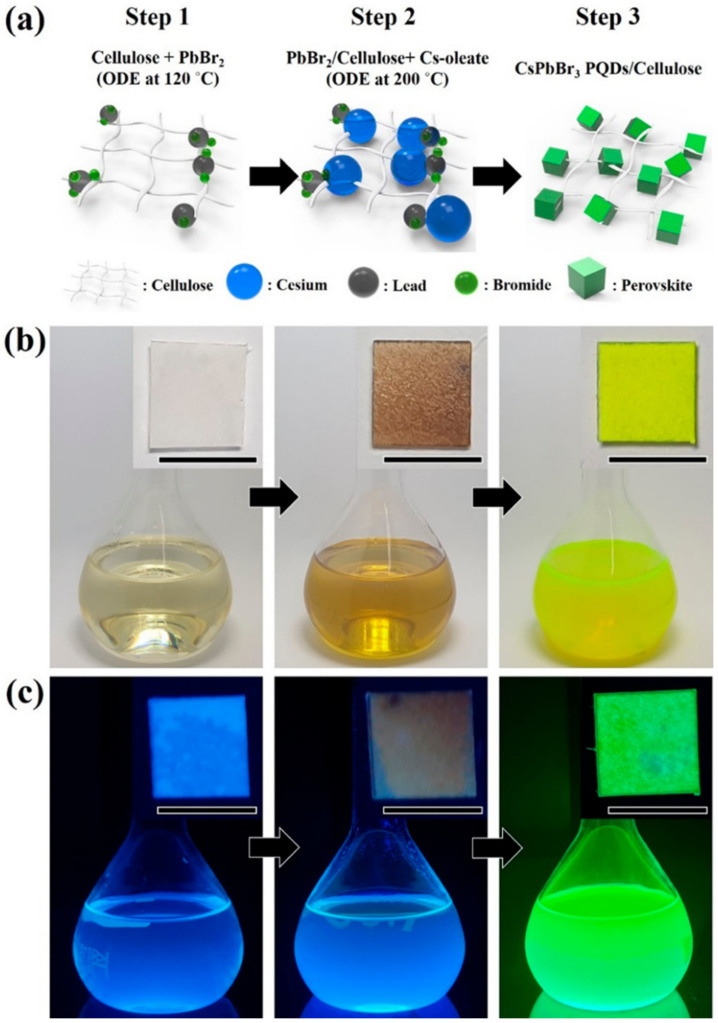
(**a**) Scheme showing the stepwise process for generating CsPbBr_3_ PQDs/cellulose composites. (**b**) Color change under visible light and (**c**) photoluminescence under UV light. Reproduced from ref. [62] Copyright (2020), with permission from American Chemical Society.

**Figure 27 polymers-15-02873-f027:**
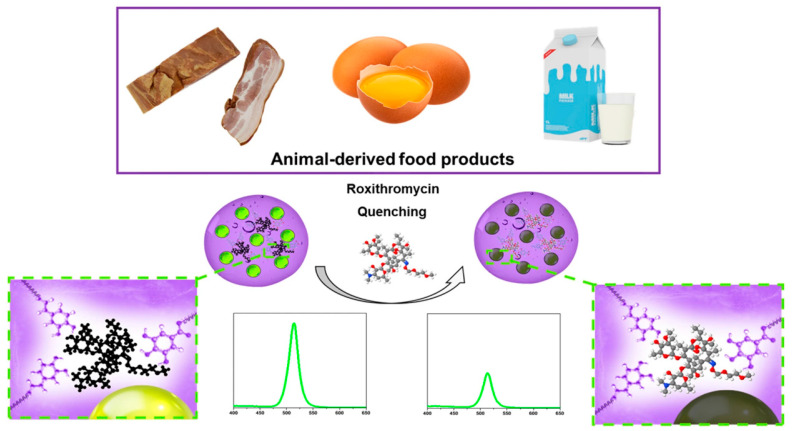
Schematic illustration of ROX sensing in animal-derived food using CsPbBr_3_-loaded MIP antioxidant-nanogels. Reproduced from ref. [63] Copyright (2022), with permission from Springer Nature.

**Figure 28 polymers-15-02873-f028:**
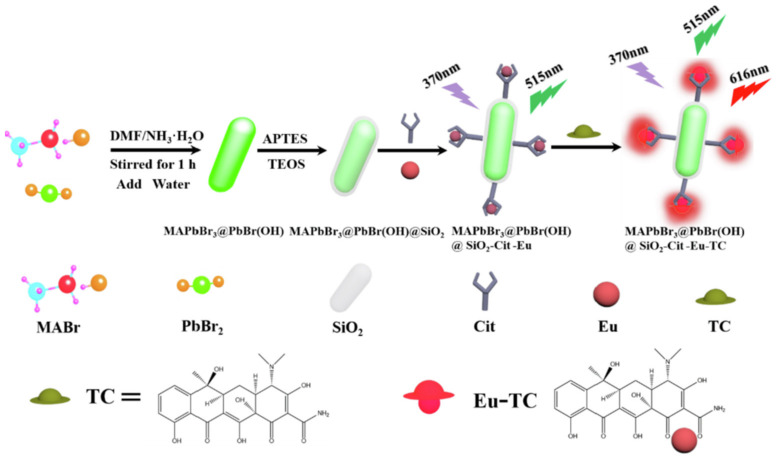
Preparation strategy of MAPbBr_3_@PbBr(OH)@SiO_2_-Cit-Eu ratio fluorescent probe and visual sensing schematic of TC. Reproduced from ref. [64] Copyright (2022), with permission from Elsevier.

**Figure 29 polymers-15-02873-f029:**
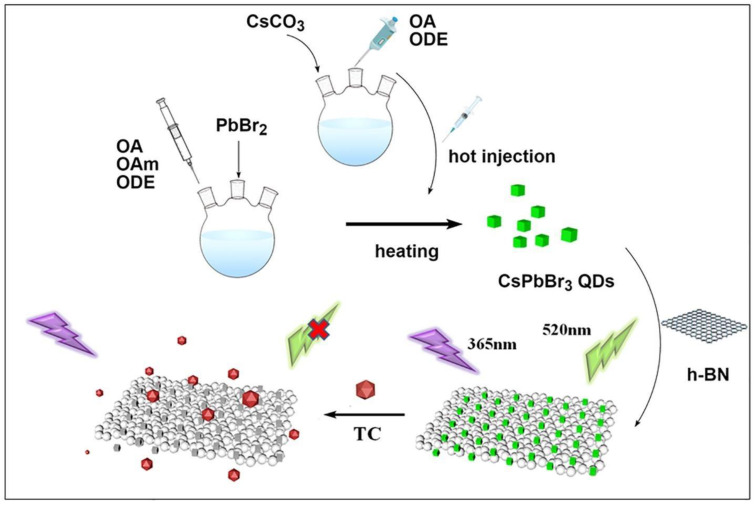
Schematic illustration of the synthesis of CsPbBr_3_@BN for sensitive detection of TC. Reproduced from ref. [66] Copyright (2021), with permission from Elsevier.

**Figure 30 polymers-15-02873-f030:**
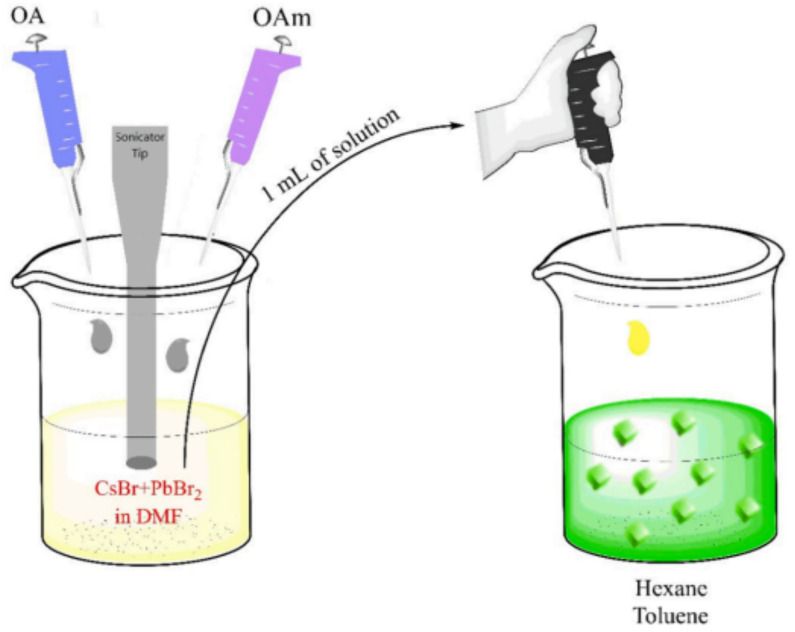
Synthesis process of CsPbBr_3_ QDs via LARP. Reproduced from ref. [68] Copyright (2023), with permission from Elsevier.

**Figure 31 polymers-15-02873-f031:**
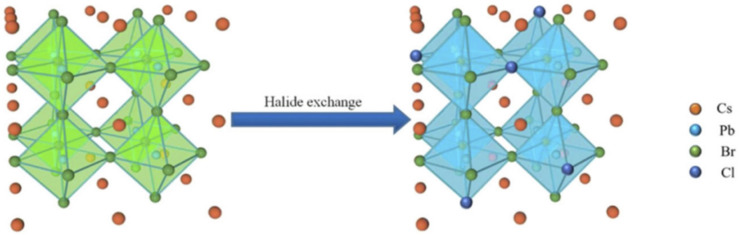
Schematic diagram of the anion exchange reaction of CsPbBr_3_ to CsPbBr_(3−x)_Cl_x_. Reproduced from ref. [69] Copyright (2022), with permission from IoP Publishing.

**Figure 32 polymers-15-02873-f032:**
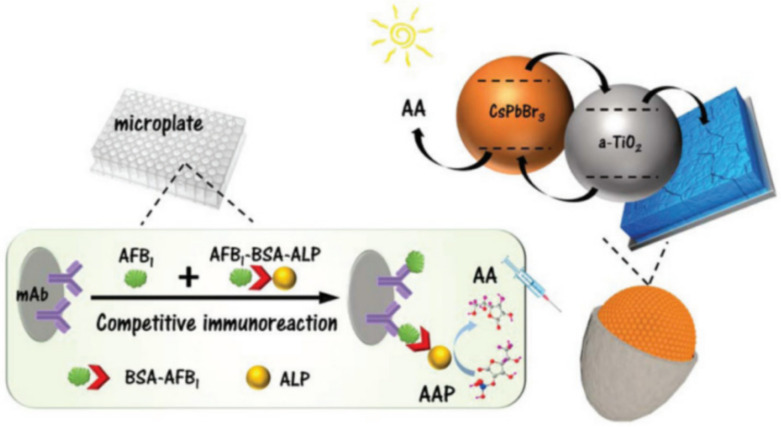
Schematic illustration of the PEC immunosensing platform for detecting AFB_1_ using CsPbBr_3_/a-TiO_2_ nanocomposites and enzyme immunoassay. Reproduced from ref. [70] Copyright (2019), with permission from Royal Society of Chemistry.

**Figure 33 polymers-15-02873-f033:**
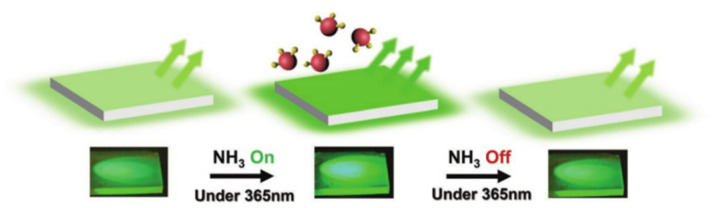
Schematic diagram of the response–recovery cycles of the dynamic passivation of PQDs. Reproduced from ref. [74] Copyright (2020), with permission from Wiley-VCH.

**Figure 34 polymers-15-02873-f034:**
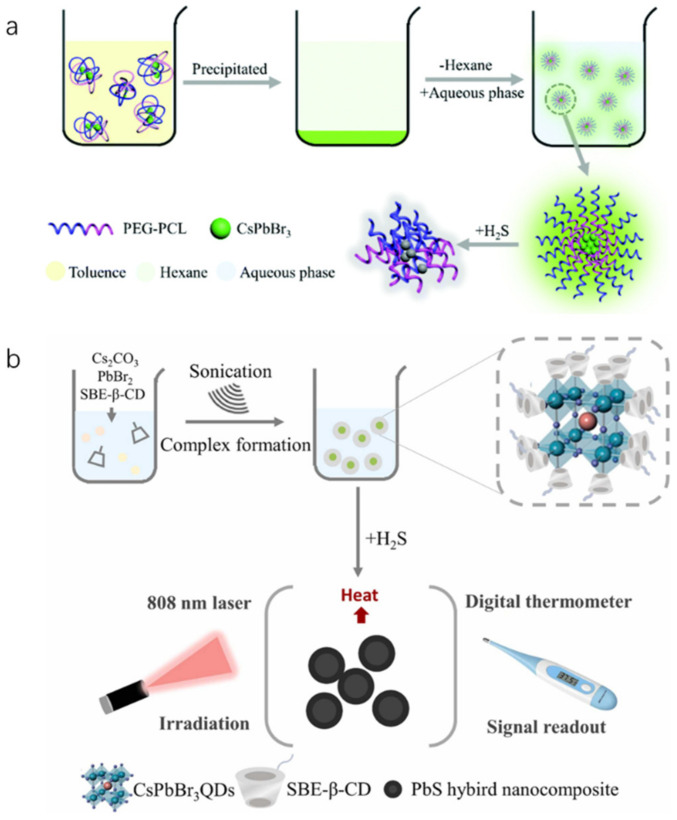
(**a**) Preparation process of CsPbBr_3_@PEG-PCL nanoparticle and the detection of H_2_S. (**b**) Preparation process of CsPbBr_3_@SBE-β-CD nanoparticles and their H_2_S photothermal sensor. Reproduced from ref. [75] Copyright (2021), with permission from Royal Society of Chemistry. Reproduced from ref. [76] Copyright (2022), with permission from Elsevier.

**Figure 35 polymers-15-02873-f035:**
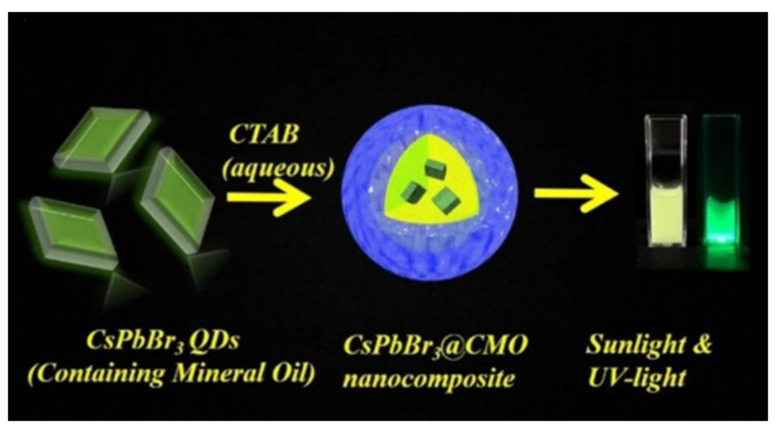
The encapsulation process of the CsPbBr_3_@CO complex.

**Figure 36 polymers-15-02873-f036:**
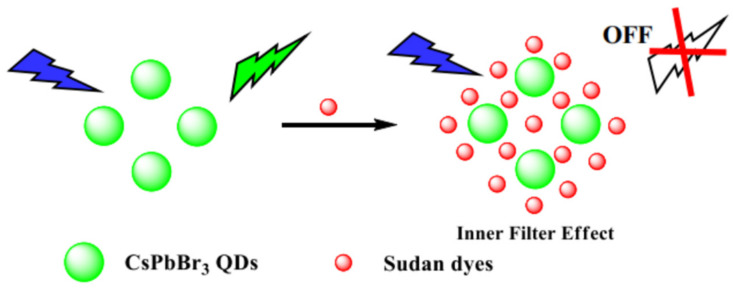
Schematic illustration of the Sudan dye detection mechanism using the CsPbBr_3_ QDs. Reproduced from ref. [78] Copyright (2018), with permission from Springer Nature.

**Figure 37 polymers-15-02873-f037:**
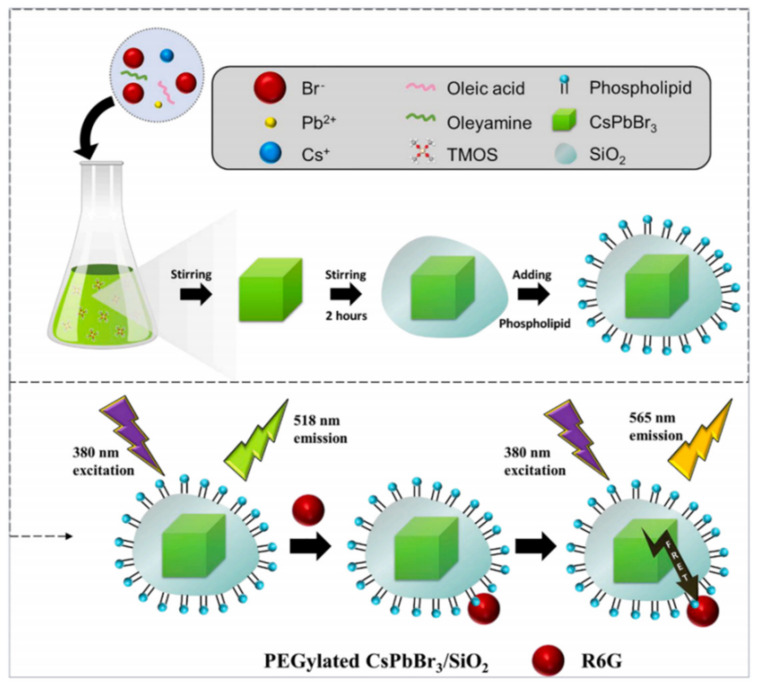
Synthesis and encapsulation of PEGylated CsPbBr_3_/SiO_2_ QDs, and FRET-based sensing mechanism of R6G using the proposed nanosensor. Reproduced from ref. [79] Copyright (2022), with permission from Elsevier.

**Figure 38 polymers-15-02873-f038:**
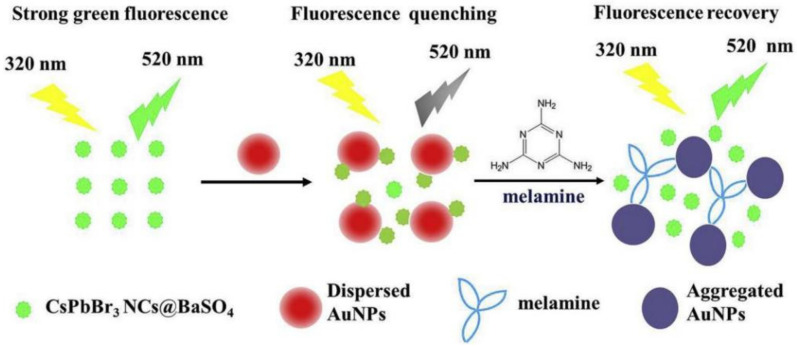
Schematic illustration of turn-on fluorescent melamine nanosensor based on the inner filter effect of the AuNPs on CsPbBr_3_ NCs@BaSO_4_. Reproduced from ref. [80] Copyright (2016), with permission from American Chemical Society.

**Figure 39 polymers-15-02873-f039:**
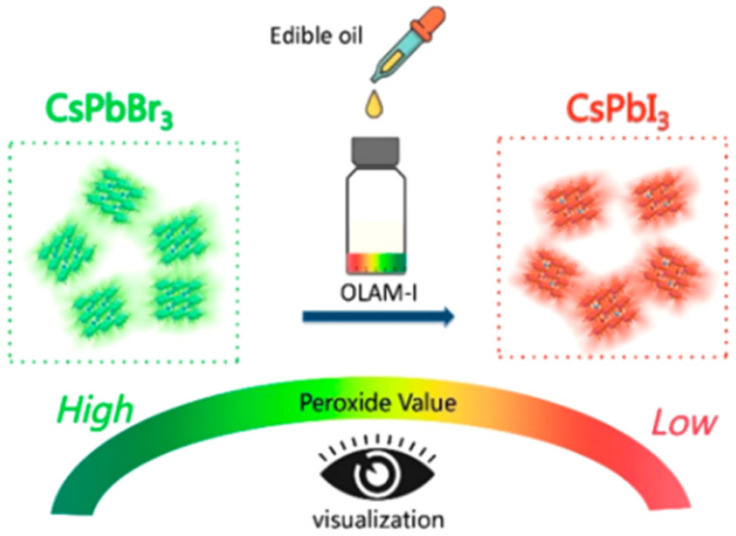
Mechanism for determination of peroxide in edible oils by using CsPbBr_3_ NCs. Reproduced from ref. [85] Copyright (2019), with permission from American Chemical Society.

**Figure 40 polymers-15-02873-f040:**
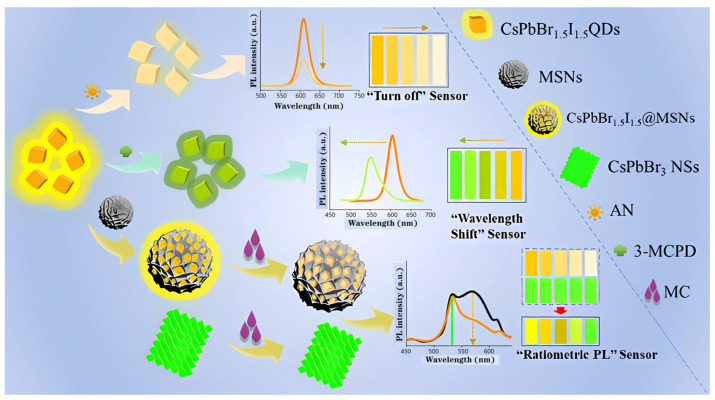
Schematic illustration of CsPbBr_1.5_I_1.5_ QDs -engineered multiplex-mode fluorescence sensing of AN, 3-MCPD, and MC in edible oil. Reproduced from ref. [86] Copyright (2021), with permission from American Chemical Society.

**Figure 41 polymers-15-02873-f041:**
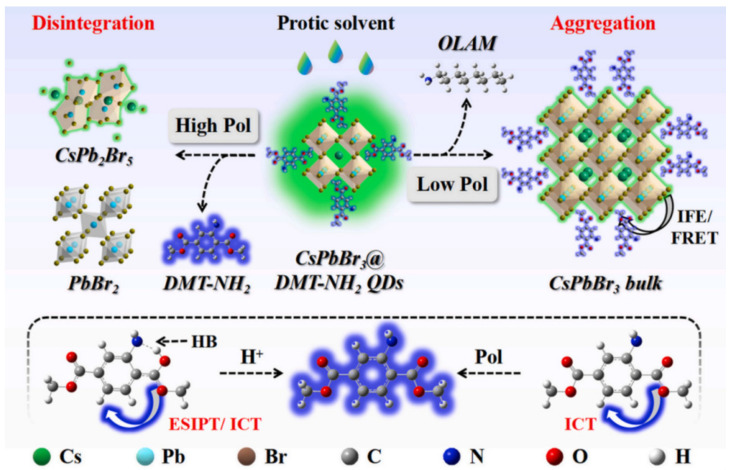
Response mechanism of CsPbBr_3_@DMT-NH_2_ QDs to water and polar protic solvents. Reproduced from ref. [87] Copyright (2022), with permission from Elsevier.

**Table 1 polymers-15-02873-t001:** Summary of methods to improve the stability of PNCs and their stability performance.

Methods	Modification Material		Water-Stabilized PNCs	Stability of PNCs	Ref.
Surface coating	Polymer	Divinylbenzene, ethyl acetate, and AIBN	CPB@SHFW	PLQY remained at 91%, 1 month in water	[19]
		PS-b-PAA	CsPbBr_3_@PS-b-PAA	PLQY remained at 60%, 2 years in water	[20]
	Inorganic oxide	C_6_H_15_O_3_Si-SH	PQDs-Pb-S-SiO_2_-SH	PLQY remained at 80%, 13 h in water	[21]
	Porous materials	Mesoporous silicon	mSiO_2_-CsPbBr_3_@AlO_x_	Over 20% PL emission intensity, 90 days in water	[22]
		Ferric organic skeleton MOFs	CsPbBr_3_@PN-333 (Fe)	The Li–O_2_ battery can be cycled stably for more than 200 h at 0.01 mA cm^−2^ under illumination	[24]
			MAPbI_3_@PCN-221(Fe_x_)	The catalytic system can operate continuously in water for more than 80 h	[25]
Metal ion doping	A site	Cs^+^ ions with FA^+^	FA_0.1_Cs_0.9_PbI_3_ PQDs	PLQY remained at 95%, several months in solution	[29]
	B site	Ni^2+^	Ni:CsPbBr_3_ PNCs	The stability of Ni:CsPbBr_3_ PNCs are higher than CsPbBr_3_ PNCs	[37]
Surface passivation	Strong chelating ligands	The polyamine chelating ligand: AHDA	AHDA-CsPbI_3_ PNCs	PLQY remained at 60%, after 15 purification cycles	[38]
		SH ligands: 2-amino-ethylmercaptan (AET)	AET-CsPbI_3_-QDs	PLQY remained at 95%, 1 h in water	[39]
	Zwitterionic organic ligands	Sulfobetaine or phosphocholine	Sulfobetaine-capped CsPbBr_3_ NCs	PLQY remained at 70–90% for 28–50 days under ambient conditions	[40]
	Organic semiconductor ligands	Rhodamine B derivative (COM)	COM-CsPbBr_3_ NCs	84% of the initial PL intensity, 300 h under HT 85 °C and HH 85%	[43]

**Table 2 polymers-15-02873-t002:** Summary of PNC-based fluorescent probes for pesticide residue detection.

Pesticide Type	Target Analyte	Molecular Structure	Fluorescent Probe	Linearity Range	LOD	Recovery Rate	Ref.
Organophosphorus	OMT	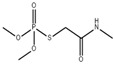	MIPs@CsPbBr_3_	50–400 ng/mL	18.8 ng/mL		[4]
	DDVP	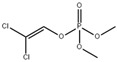	CsPbBr_3_ QDs	5–25 μg/L	1.27 μg/L	87.4–101%	[46]
	Phoxim	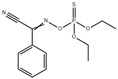	CsPbBr_3_ QDs	5–100 ng/mL	1.45 ng/mL		[47]
Triazine herbicides	Prometryn	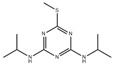	MIP/CsPbBr_3_- QDs		0.010 μg/kg, 0.050 μg/L	88.0–106.0%	[49]
	Simazine	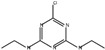	MIP-CsPbBr_3_	0.1–500.0 μg/L	0.06 μg/L	86.5–103.9%	[50]
Organochlorine	OCPs		MAPB-QDs				[51]
Other pesticides	Clodinafop	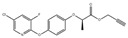	CsPbI_3_ PQDs	0.1–5.0 μM	34.70 nM	97–100%	[1]
	Propanil	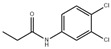	MIP-QDs	1.0 μg/L– 2 × 10^4^ μg/L	0.42 μg/kg, 0.38 μg/L	87.2–112.2%	[52]

**Table 3 polymers-15-02873-t003:** Summary of PNC-based fluorescent probes for environmental pollutant detection.

Target Analyte	Molecular Structure	Fluorescent Probe	LinearityRange	LOD	Ref.
U	U	CsPbBr_3_ PQD	0–3300 nM (3.3 μM)	0–3300 nM (3.3 μM)	[53]
O-nitrophenol (ONP)	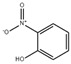	CsPbBr_3_@SHFW	0–280 μM	7.69 × 10^−3^ μM	[54]

**Table 4 polymers-15-02873-t004:** Summary of PNC-based fluorescent probes for ion detection.

Ion Type	Target Analyte	Fluorescent Probe	Linearity Range	LOD	Ref.
Cation	Cu^2+^	CsPbBr_3_ (PQD)	0–100 nM	0.1 nM	[55]
	Cu^2+^	CsPbBr_3_ (CPB)	10^6^ M–10^2^ M		[56]
	Hg^2+^	CH_3_NH_3_PbBr_3_ (QDs)	0–100 nM	0.124 nM (24.87 ppt)	[57]
	Fe(III)	CPB@PSAA	5–150 μM	2.2 μM	[58]
	Zn^2+^, Ag^+^	CQD-MAPbBr_3_@SiO_2_			[59]
	Pb^2+^	CH_3_NH_3_PbBr_3_ (MAPbBr_3_)			[60]
	Pb^2+^	MAPbBr_3_		5 × 10^−3^ μg/mL	[61]
Anion	I^−^, Cl^−^	CsPbBr_3_ PQDs		2.56 mM, 4.11 mM	[62]

**Table 5 polymers-15-02873-t005:** Summary of PNC-based fluorescent probes for antibiotic detection in food.

Target Analyte	Molecular Structure of the Target Analyte	Fluorescent Probe	Linearity Range	LOD	Recovery Rate	Ref.
Roxithromycin	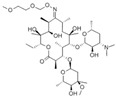	MIP-CsPbBr_3_		1.7 × 10^5^μg/mL (20.6 pM)		[63]
Tetracycline	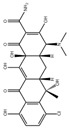	MAPbBr_3_ @PbBr(OH)@SiO_2_-Cit-Eu	0–25 μM	11.15 nM		[64]
		Cs_4_PbBr_6_/CsPbBr_3_	0.4–10 μM	76 nM		[65]
		CsPbBr_3_@BN	0–0.44 mg/L	6.5 μg/L		[66]
		APTES@IPQDs				[67]
Cefazolin	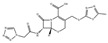	CsPbBr_3_ QDs	25–300 nM	9.6 nM	94–106%	[68]
Ciprofloxacin hydrochloride	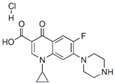	CsPbBr_(3−*x*)_Cl*_x_* NCs				[69]

**Table 6 polymers-15-02873-t006:** Summary of PNC-based fluorescent probes for the detection of microbial toxins, pathogens, and carcinogens in food.

Target Analyte	Molecular Structure	Fluorescent Probe	Linearity Range	LOD	Ref.
B_1_ (AFB_1_)	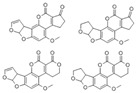	CsPbBr_3_/a-TiO_2_	0.01~15 ng/mL	2.8 pg/mL	[70]
MAPB QDs@SiO_2_ and MAPB		8.5 fg/mL	[71]
SEs	Staphylococcal enterotoxin	CsPb_2_Br_5_ @MSN			[72]
Artificial wax on fruit	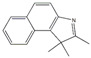	Cs_2_PdBr_6_			[73]

**Table 7 polymers-15-02873-t007:** Summary of PNC-based fluorescent probes for the detection of food spoilage gas in food.

Target Analyte	Fluorescent Probe	Linearity Range	LOD	Ref.
NH_3_	CsPbBr_3_ QDs	25–350 ppm	8.85 ppm	[74]
H_2_S	CsPbBr_3_@PEG-PCL			[75]
H_2_S	CsPbBr_3_@SBE-β-CD	0.5–6000.0 μM	0.19 μM	[76]
H_2_S	CsPbBr_3_@CO	0.15–105.0 μM	53.0 nM	[77]

**Table 8 polymers-15-02873-t008:** Summary of PNC-based fluorescent probes for the detection of harmful additives in food.

Target Analyte	Molecular Structure	Fluorescent Probe	LinearityRange	LOD	Recovery Rate	Ref.
Sudan red I	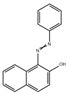	CsPbX_3_	0.5–150 μg/L	0.3 μg/L	95.27–105.96%	[2]
Sudan red I–IV	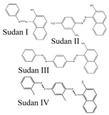	CsPbBr_3_		3.33, 0.03, 0.03, 0.04 ng/mL		[78]
Rhodamine 6G	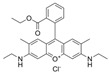	CsPbBr_3_/SiO_2_ QDs	0–10 mg/mL	0.01 mg/mL		[79]
CPBQDs/PSFM		0.01 ppm		[80]
RhoB	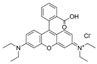	CsPbBr_3_-PVDF		0.01 ppm		[81]
Basic yellow dye	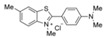	CsPbBrI_2_ QDs	1–500 μg/mL	0.78 μg/mL	95.27–98.84%	[82]
Melamine	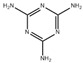	CsPbBr_3_NCs@BaSO_4_		0.42 nmol/L		[83]
Bisphenol A	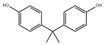	Cs_4_PbBr_6_@CsPbBr_3_ PQD				[84]

**Table 9 polymers-15-02873-t009:** Summary of PNC-based fluorescent probes for edible oil quality inspection.

Target Analyte	Molecular Structure	Fluorescent Probe	LinearityRange	LOD	Recovery Rate	Ref.
Peroxide value	CH_3_(CH_2_)_7_CH=CH(CH_2_)_7_CH_2_NH_3_I	CsPbBr_3_ NCs				[85]
AN,3-MCPD,MC	Excessive acid number 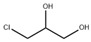 , H_2_O	CsPbBr_1.5_I_1.5_@MSNs		0.71 mg KOH/g, 39.8 μg/mL 3-MCPD, 0.45% MC		[86]
H_2_O	H_2_O	CsPbBr_3_@DMT-NH_2_QDs		0.006% (*v*/*v*), 0.01% (*v*/*v*)	93.0~108.0%	[87]
TPM	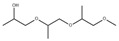	CsPbBr_3_ QD	17–31.5%, 25–31.5%, 21.5–33%			[88]

## Data Availability

Not applicable.

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
