# Peer review of "Application of Perovskite Nanocrystals as Fluorescent Probes in the Detection of Agriculture- and Food-Related Hazardous Substances"

_polymers, 2023, doi:10.3390/polym15132873_

Round 1
Reviewer 1 Report
In this manuscript, the authors briefly introduce the structure and properties of halide perovskite nanocrystals, and summarize methods of PNCs stabilization including surface coating, ion doping, surface passivation. They reviewed the fluorescent probe function of PNCs for detection of pesticide residue, environmental pollutant, cation and anion ions, antibiotics, and harmful additives in agriculture and food field. The mechanism and detection sensitivity were demonstrated very detailly. Finally, they summarized the challenges of PNCs as detection prob in agriculture and food fields and prospected the future. This review presents us very rich content. However, there are some concerns need to be addressed before acceptance.
Please correct ‘It’ to lower case in penultimate line of abstract.
Please add reference citation and copyright permission of each figure, although the authors cite the reference in the description paragraph.
There are too many figures in the review, please re-organize the figures and meet the figure number requirement of this journal.
Please confirm position of reference 37 and 38 citation.
There is lack of reference citation in section 2.1.1, please add the citation
Please correct ‘20.0x103’ to superscript of ‘3’ on page 16
Please uniform the font style on page 33, which is different with all others.
Moderate editing of English language required
Author Response
Dear reviewer:
Thank you very much for your time and useful advices which are very helpful and improve the quality of our manuscript. We hope that the following responses address your questions and suggestions. Please find it in the attachment.
Best wishes,
Tengling Ye

Reviewer 2 Report
Application of perovskite nanocrystals as fluorescent probes in the detection of agriculture- and food-related hazardous substances
Manuscript ID: polymers - 2417128
Comments to the Authors
This paper reviews the potential of halide perovskite nanocrystals (PNCs) as a new kind of luminescent materials for fluorescent probes for detecting hazardous substances in food and agriculture. The paper summarizes the water stabilization methods such as polymer surface coating, ion doping and surface passivation. The recent advances of PNCs as fluorescent probes for detecting hazardous substances in the field, and the detection effect and mechanism are discussed and analyzed. Finally, the problems and solutions faced by PNCs as fluorescent probes in agriculture and food are also summarized and prospected.
The work is significant because it provides a comprehensive overview of the synthesis of halide PNCs and their applications in detection of food and agriculture related hazardous substances. They also emphasized on how the PNCs can be made more stabilized in different environments such as in water.
I recommend the manuscript for publication and sharing a few comments and suggestions below for some clarifications and better readability:
1. The authors have provided a good background on the motivation behind this work. Can they also provide some discussion on the long-term effect on toxicity of halide PNCs in agriculture and food related substances?
2. The authors have provided clear and comprehensive figures. Can they also provide more descriptive figure legends? Providing information on permissions of figure use and references will be helpful.
3. The tabulated summaries of applications of halide PNCs for detection of different substances is very useful. A similar tabulated summary of different synthesis methods and their advantages and disadvantages might also be very useful.
4. The discussion on the different halide PNCs is detailed and useful. I feel adding some discussion on the challenges involved in characterizing these materials might also be useful.
Author Response
Dear reviewer:
Thank you very much for your time and useful advices which are very helpful and improve the quality of our manuscript. We hope that the following responses address your questions and suggestions. Please find it in the attachment.
Best wishes,
Wei Zhao
